# Abrupt increase in Arctic-Subarctic wildfires caused by future permafrost thaw

In-Won Kim [1,2] ✉, Axel Timmermann [1,2], Ji-Eun Kim [1,2], Keith B. Rodgers [3], Sun-Seon Lee [1,2], Hanna Lee [4] & William R. Wieder [5,6]

Unabated 21st-century climate change will accelerate Arctic-Subarctic permafrost thaw which can intensify microbial degradation of carbon-rich soils, methane emissions, and global warming. The impact of permafrost thaw on future Arctic-Subarctic wildfires and the associated release of greenhouse gases and aerosols is less well understood. Here we present a comprehensive analysis of the effect of future permafrost thaw on land surface processes in the Arctic-Subarctic region using the CESM2 large ensemble forced by the SSP3-7.0 greenhouse gas emission scenario. Analyzing 50 greenhouse warming simulations, which capture the coupling between permafrost, hydrology, and atmosphere, we find that projected rapid permafrost thaw leads to massive soil drying, surface warming, and reduction of relative humidity over the Arctic-Subarctic region. These combined processes lead to nonlinear late-21st-century regime shifts in the coupled soil-hydrology system and rapid intensification of wildfires in western Siberia and Canada.

The cold environment of the Arctic is typically associated with long climatological fire return intervals. However, anomalously warm and dry summers can create conditions that promote wildfires, such as those in northeastern Siberia in 2020–21[1]. Fires that occur in the carbon-rich soils of the Arctic and Subarctic can last for months, releasing large amounts of aerosols and carbon[1–3], with potential impacts on the Earth's radiation budget and climate. Fire occurrences in the Arctic and Subarctic are in part controlled by the immediate atmospheric conditions (fire weather)[4], and in part by soil water conditions[5], with arid soils increasing the frequency, and spatial extent of fires, and by the availability of fire fuel. Changes in high-latitude soil moisture are determined by the imbalance between precipitation, evapotranspiration, snow- and ice-melt, as well as runoff[6,7]. These processes act on different timescales giving rise to complex variations in soil moisture and, in turn, fire occurrences. In the Arctic and Subarctic regions, runoff is influenced by the presence or absence of deep soil permafrost which can act as a barrier preventing the drainage of liquid water from the upper soil layers[8–10].

As a result of recent Arctic warming, permafrost in some regions has already begun to thaw, gradually deepening the soil active layer and initiating changes in hydrological processes and soil moisture content[11,12]. In some specific areas (e.g., North slope Alaska, Hudson Bay lowlands of Canada, and Dmitri Laptev strait, etc.) abrupt shifts in permafrost have been observed[13], which can cause ground subsidence and corresponding hydrological changes[14,15].

Capturing these processes and projecting their future sensitivity as well as impacts on wildfires in Earth system models (ESMs) has remained a major challenge[16–18], in part due to the range of spatial scales of interactive processes contributing to permafrost dynamics[12,19]. So far, 13 models out of 134 ESMs participating in the recent Coupled Model Intercomparison Project, version 6 (CMIP6)[20] have represented the coupling between permafrost, soil hydrology, and fires[18,21]. A previous study[22] identified abrupt increases in potential fire severity following future permafrost degradation. That study used estimates of the Fire Weather Index (FWI), which translates atmospheric conditions to fire risk without accounting explicitly for changes in vegetation, fire fuel, or soil hydrology.

However, to date, a fully coupled assessment of future wildfire, permafrost, and soil hydrology interactions has not been conducted.

[1]Center for Climate Physics, Institute for Basic Science, Busan, South Korea. [2]Pusan National University, Busan, South Korea. [3]WPI-Advanced Institute for Marine Ecosystem Change, Tohoku University, Sendai, Japan. [4]Norwegian University of Science and Technology, Trondheim, Norway. [5]Climate and Global Dynamics Laboratory, National Center for Atmospheric Research, Boulder, CO, USA. [6]Institute of Arctic and Alpine Research, University of Colorado Boulder, Boulder, CO, USA. ✉e-mail: iwkimi@pusan.ac.kr

Here we set out to study the impact of rapid permafrost thaw on high latitude wildfires using the Community Earth System Model 2 (CESM2) large ensemble (CESM2-LE)[23] forced under a historical/Shared Socioeconomic Pathways (SSP) 3-7.0 (see Methods). Our focus will be to elucidate the mechanisms that trigger abrupt regime shifts in fire activity.

## Results

### Rapid permafrost thaw

To provide an overall empirical context for forced changes in permafrost, we compare estimates of observed linear trends in 2 m air temperature (T2M), ground temperature (TG), and active layer thickness (ALT) for the period 1997–2019 with corresponding simulated trends obtained from the 50 ensemble members of the CESM2-LE (Fig. S1) for the same period. The ALT has increased over the permafrost regions with Arctic warming, from 1997 to 2019 (Fig. S1a–c). The increasing trends in ALT are particularly pronounced along the southern margin of the permafrost zone (Fig. S1c), where the thaw threshold will be crossed more easily due to higher climatological summer temperatures as well as in northwestern Siberia [60–80°E, 65–70°N]. Most ensemble members in the CESM2-LE over this region show increasing trends in the T2M, TG, and ALT, but the trends are on average weaker

than those from the model-derived reanalysis (Fig. S1d–f), which suggests that the observed trends tend to be much higher than natural variability, as represented by the CESM2-LE ensemble spread. The advantage of using a large ensemble is that it allows us to discern forced changes from natural variability, as every time point of the ensemble has 50 realizations of natural climate variability. Comparing natural variability with projected trends from 1997 to 2019 in the T2M, TG, and ALT reveals that the anthropogenic greenhouse warming (as represented by the ensemble mean) in the region under consideration is already emergent above the internal variability (Fig. S1d–f), at least beyond the interquartile range.

To identify the timing of major changes in projected soil variables, we conducted for each ensemble member a change point analysis[24,25] of ALT, soil ice content, and soil moisture (see Methods). The resulting patterns (Fig. 1a, c, e) illustrate the timing of rapid forced shifts in soil properties. Rapid changes in ALT and soil ice content mainly emerge over western Siberia, far eastern Siberia, and Canada from the mid-to-late 21st century. The timing of abrupt changes in ALT and soil ice content tends to be similar (Fig. 1a, c). Furthermore, a substantial increase in ALT is observed in western Siberia, far eastern Siberia, and Canada in comparison to the historical period, which reflects changes in the forced response of the permafrost column (Fig. 1b). In these areas,

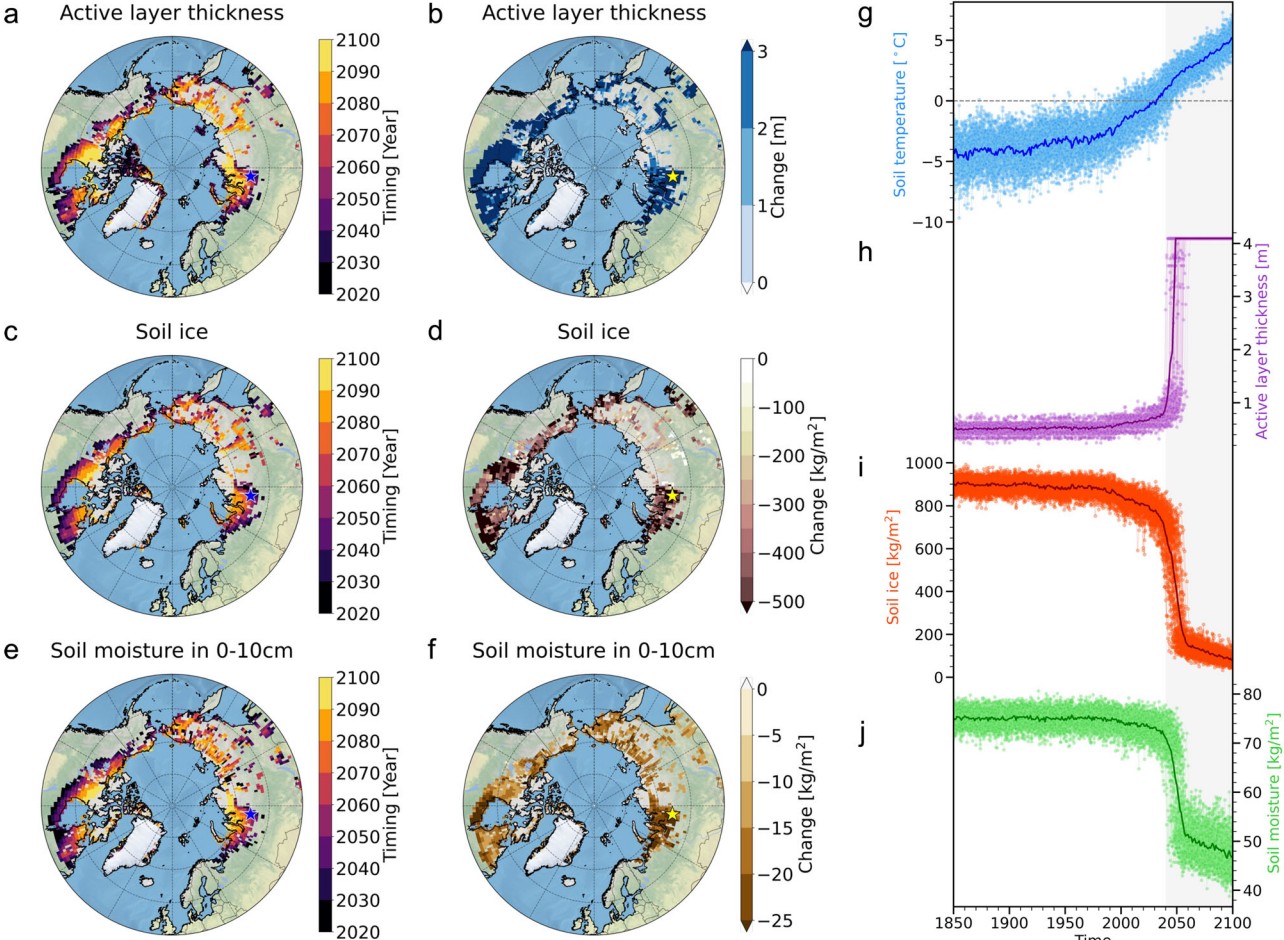

**Fig. 1 | Abrupt changes over the historical permafrost regions.** The timing of abrupt changes in (**a**) maximum annual active layer thickness (ALT), (**c**) soil ice content, and (**e**) soil moisture in 0–10 cm depth, which is defined by a median year among abrupt changes from the 50 ensemble members (units: year). The abrupt changes of (**b**) ALT (units: m), (**d**) soil ice content (units: kg/m²), and (**f**) soil moisture in 0–10 cm depth (units: kg/m²), which is defined by differences during 20 years of pre- and post- abruptness. Blue (or yellow) star markers in panels **a**–**f** indicate a representative grid box in western Siberia (65.5°N, 83.75°E). Time evolution of (**g**)

soil temperature in 0–10 cm depth (units: °C) (blue), (**h**) ALT (units: m) (purple) for an exemplary grid point in the representative grid box (65.5°N, 83.75°E), (**i**) soil ice content (units: kg/m²) (red), and (**j**) soil moisture in 0–10 cm depth (units: kg/m²) (green) in 50 ensemble members. Bold lines indicate ensemble means and thin lines indicate individual ensemble members in panels **g**–**j**. Here we focus on near-surface permafrost processes. We therefore define the historical permafrost regions as the area where ALT is less than 3 m for the period of 1850–1869[54–56].

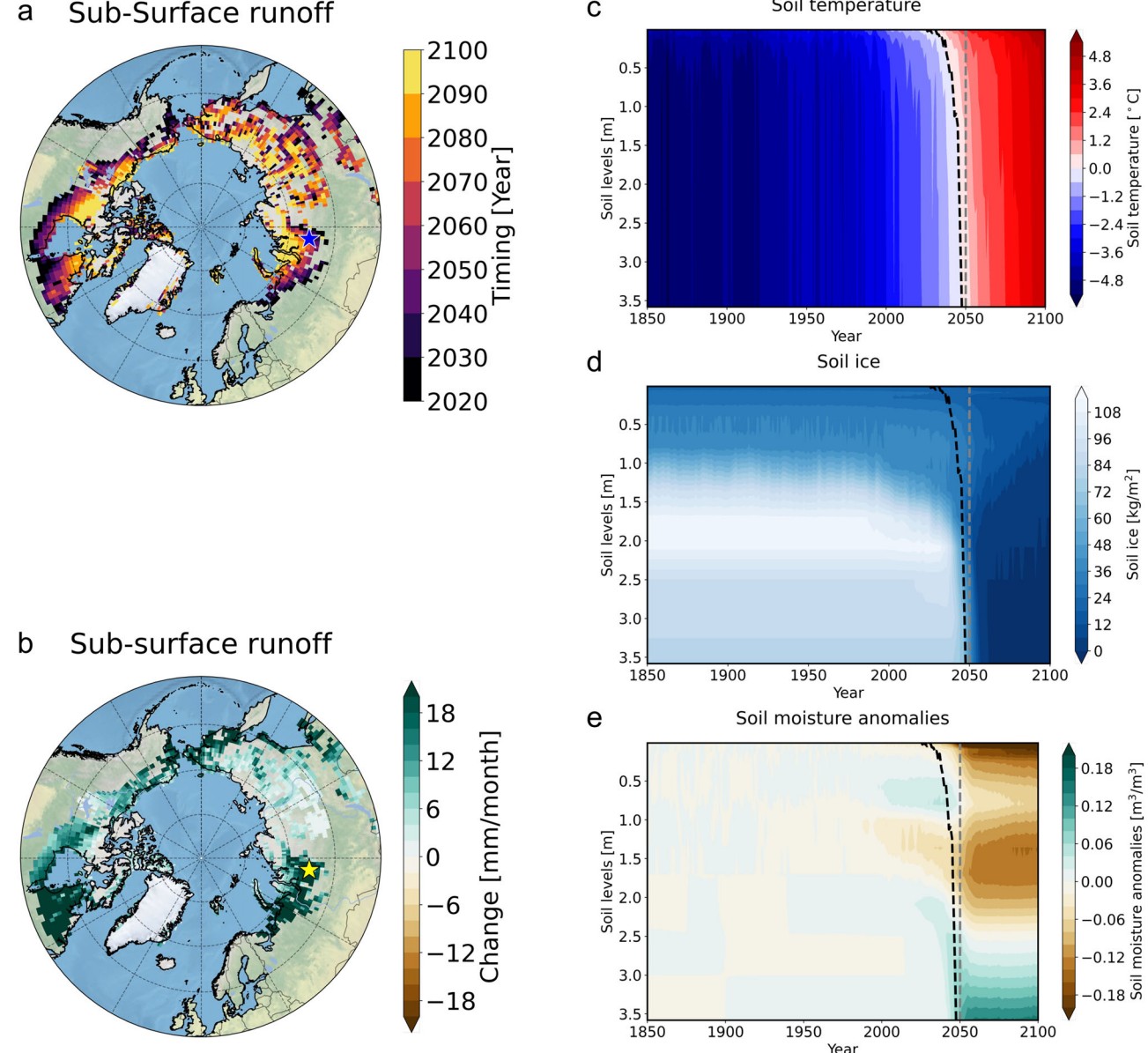

**Fig. 2 | Abrupt change in thermal and hydraulic properties over the historical permafrost regions. a** The timing of the abrupt change in sub-surface runoff (units: year), (**b**) the magnitude of abrupt change in sub-surface runoff (units: mm/month) during 20 years of pre- and post- abruptness, and (**c**–**e**) vertical time cross-sections in western Siberia (65.5°N, 83.75°E) in 50 ensemble mean: Blue (or yellow)

star markers in panels **a**, **b** indicate a representative area in western Siberia (65.5°N, 83.75°E). **c** soil temperature (units: °C), (**d**) soil ice content (units: kg/m²), and (**e**) volumetric soil water anomalies relative to 1850–1950 (units: m³/m³). Black dashed lines in panels **c**–**e** indicate the time when the soil temperature in each layer reaches 0 °C.

the soil ice content also decreases by more than about 300 kg/m² during the 20-year pre- and post-thaw periods (Fig. 1d). For illustrative purposes, we focus here on time series of individual ensemble members over a grid box in western Siberia [65.5°N, 83.75°E] (yellow star in Fig. 1b, d, f). The region exhibits a clear and abrupt increase in ALT (2040: 0.93 m, 2060: 4.09 m) and a reduction in soil ice (2040: 668 kg/m², 2060: 149 kg/m²) over the period 2040–2060 (Fig. 1h, i). This extremely abrupt threshold response stands in stark contrast to the much more gradual warming at 10 cm soil temperature (Fig. 1g). In addition, the timing of the 0 °C soil temperature in 0–10 cm depth exceedance serves as a good proxy for the abrupt responses in soil ice in this region.

**Hydrological responses to rapid permafrost thaw**
Our simulations also show a rapid mid-to-late 21st-century decrease in upper soil moisture and an increase in subsurface runoff over western

Siberia, far eastern Siberia, and Canada (Figs. 1e, f, and 2a, b). Notably, this coincides with a rapid decrease in soil ice over the same regions (Fig. 1c, d) that have relatively deep soils and higher soil moisture content (Figs. 1f, and S2a, b).

Due to the spatial heterogeneity of permafrost thaw and soil moisture, regionally averaged datasets can obscure a more accurate understanding of the mechanisms sustaining abrupt soil drying following the permafrost thaw. Therefore, to elucidate the underlying mechanisms of soil hydrological changes in response to rapid permafrost thaw, we focus our initial analysis, as previously described, on a single grid cell in western Siberia. In this region, near-surface soil moisture rapidly declines by 28% (2040: 71.1 ± 2.9 kg/m², 2060: 51.4 ± 3.5 kg/m²) (Fig. 1j), synchronous with the timing of rapid permafrost thaw (Fig. 1h, i). The vertical soil profiles at this location reveal that at the time of the abrupt

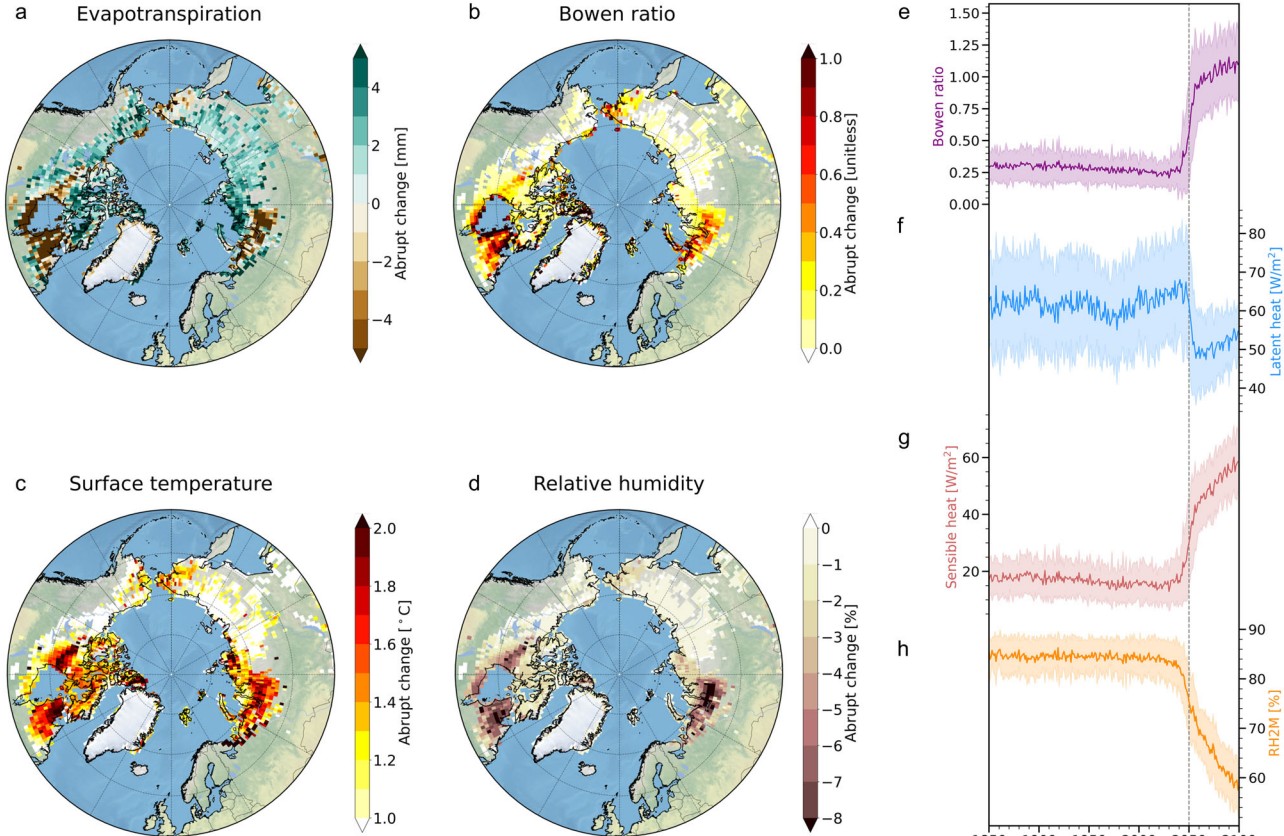

**Fig. 3 | Changes in near-surface atmospheric conditions in July and August following rapid permafrost thaw over the historical permafrost regions.** The differences during 20 years of pre- and post-thaw (calculated for each grid point) in (**a**) evapotranspiration (units: mm), (**b**) Bowen ratio (units: unitless), (**c**) surface air temperature (units: °C), and (**d**) relative humidity at 2 m (units: %) in July and August. Time evolution of (**e**) Bowen ratio (units: unitless), (**f**) latent heat flux (units: W/m²), (**g**) sensible heat flux from land to the atmosphere (units: W/m²), and (**h**) relative humidity at 2 m (units: %) in western Siberia (65.5°N, 83.75°E) in July. Bold lines indicate ensemble means, and shading indicates ±1 standard deviation of ensemble members in panels **e–h**.

transition, soil temperatures in the upper layers reach 0 °C around 2030. The warming then propagates to deeper layers (1–3 m), reaching 0 °C by ~2050 through the downward heat transfer (Fig. 2c). Subsequently, soil ice in the deeper soil layers melts away around 2050 (Fig. 2d). After the rapid soil ice melting, soil moisture in the upper layer percolates into the deeper soil layers due to an increase in vertical hydraulic conductivity[26] in the deeper soil layer in the model. In turn, upper soil moisture decreases (0–2 m) and deeper soil moisture (>3 m) increases abruptly (Fig. 2e). The runoff from the surface, surface water storage, and sub-surface show sudden changes after the thaw (Fig. S3a–c). The steep decrease in surface runoff from surface and surface water storage after 2050 results from soil moisture depletion (Fig. S3b, c). In contrast, precipitation increases monotonically after 1980, and abrupt changes in rainfall and snowfall are not evident during the time of rapid permafrost thaw (Fig. S3d, e). Transpiration and evaporation from the canopy increase by approximately a factor of 2.6 relative to the pre-thaw period (Fig. S3f, g). However, the magnitudes of abrupt changes in canopy evapotranspiration are approximately half those of the ground evaporation changes associated with sparse vegetation across these regions (Fig. S3f–h).

The temporal evolutions in other permafrost locations show that a rapid decrease in soil ice mainly occurs in ice-rich areas, which is consistent with sudden shifts in sub-surface runoff and soil moisture from the mid-to-end of the 21st century (Fig S4). In contrast, in regions with less soil ice, the reduction in soil ice occurs more gradually after the early 21st century, thereby sustaining sub-surface runoff and upper soil moisture levels (Fig. S5).

## Land-atmosphere interactions caused by abrupt soil drying
In general, soil moisture anomalies generate changes in surface energy exchange via land-atmosphere interactions, thereby modulating near-surface atmospheric conditions[27]. A specific case was analyzed through field experiments in northeastern Siberia to understand the effects of drainage on the energy balance in permafrost regions[28]. The study showed that a drainage-induced decrease in soil moisture increased (decreased) sensible heat fluxes (latent heat fluxes) in summer[28]. Following this line of argument, we use the CESM2-LE to further document how the summer atmosphere responds to abrupt soil drying.

Following rapid permafrost thaw, a distinct decrease in total evapotranspiration is prevalent along with an increase in the Bowen ratio (the ratio of sensible heat flux to latent heat flux) in July and August over Canada [55–60°N, 60–90°W] and western Siberia [65–70°N, 60–90°E] (Fig. 3a, b). In addition, an increase in surface air temperature of greater than 2 °C and a decrease in relative humidity are manifested over these regions (Fig. 3c, d), which indicates that the abrupt soil drying following the permafrost thaw alters local atmospheric conditions in summer.

To further elucidate interactions between the land and atmosphere induced by abrupt soil drying that occurs after the thaw, we analyze the time evolution of the surface energy budget in July over a representative location in western Siberia [65.5°N, 83.75°E]. After the abrupt soil drying, the Bowen ratio increases abruptly around 2050 (Fig. 3e). The latent heat flux abruptly decreases due to the loss of evapotranspiration (2040: 66.9 ± 14.1 W/m², 2060: 50.7 ± 10.3 W/m²), accompanying a rapid increase in the sensible heat flux (2040: 16.8 ± 8.7 W/m², 2060: 44.4 ± 10.4 W/m²) (Fig. 3f, g). Ground heat fluxes into the soil also rapidly

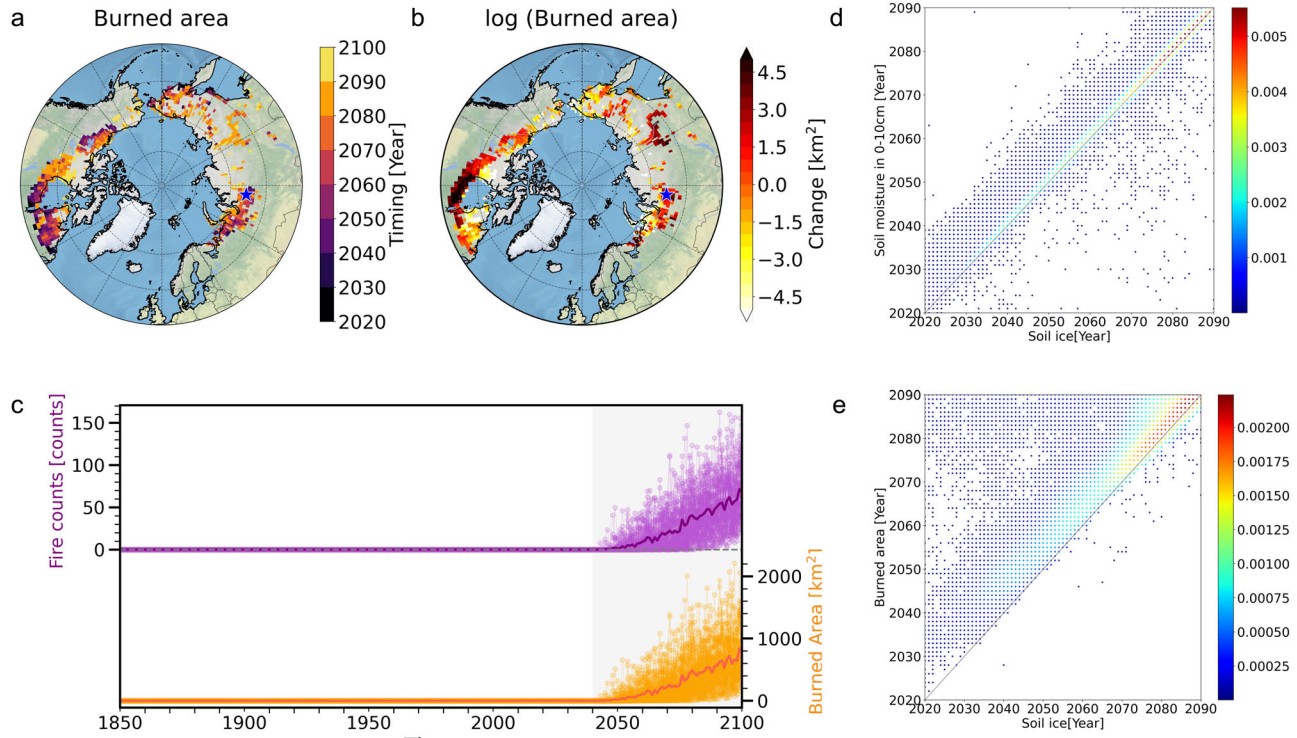

**Fig. 4 | Abrupt changes in the burned area over the historical permafrost regions. a** The timing of abruptness in the burned area (units: year), (**b**) the difference in the logarithm of the burned area [log (burned area)] during 20 years of pre- and post-abruptness (units: km²): Blue star markers in panels **a**, **b** indicate a representative grid box in western Siberia (65.5°N, 83.75°E), (**c**) temporal evolution of fire counts (units: counts) and burned area (units: km²) over a representative grid box in western Siberia (65.5°N, 83.75°E). Bold lines indicate the ensemble mean and thin lines indicate individual ensemble members in panel **c**. **d**, **e** Relationship between the timing of permafrost thaw and abrupt changes in soil moisture and burned area over the historical permafrost regions (units: year): (**d**) soil ice content and soil moisture in 0–10 cm depth, and (**e**) soil ice content and burned area in the 50 ensemble members (units: year). Red indicates a higher probability density of grid points and blue a lower probability density in panels **d**, **e**.

decrease after the abrupt soil drying (2040: 28.4 ± 5.8 W/m², 2060: 21.3 ± 5.1 W/m²), as dry soil has a lower heat capacity and thermal conductivity relative to wet soil[28–30] (Fig. S6a). Along with these changes, the ensemble mean of surface air temperature increases considerably (2040: 15.4 ± 3.0 °C, 2060: 18.5 ± 2.8 °C) (Fig. S6b). Once the sensible heat flux increases abruptly following the abrupt soil drying, this can further accelerate an increase in surface air temperature, thus leading to a rapid decline in relative humidity (2040: 82.5 ± 5.1%, 2060: 68.6 ± 4.8%), despite a smaller change in the actual amount of water vapor (Figs. 3g, h, and S6b, c). Furthermore, drier atmospheric conditions can increase atmospheric water demand, leading to an increase in canopy evapotranspiration[27,31]. Additionally, the effect of future CO₂ fertilization can enhance vegetation growth, which further amplifies evapotranspiration. The rapid increase in canopy evapotranspiration mainly occurs over areas where soil ice melts quickly, which is consistent with the timing of abrupt soil drying and atmospheric drying (Figs. 3, and S3–5).

### Wildfire responses to abrupt soil drying over the historical permafrost regions

The abrupt soil drying and intensified atmospheric aridity can facilitate an abrupt increase in fires, related to biomass and peat burning over the permafrost regions. Abrupt increases in burned areas are pronounced over the historical permafrost regions (Fig. 4). The burned area after the rapid permafrost thaw is ~2.6 times greater than that observed during the pre-thaw period (Fig. S7). Over western Siberia [65.5°N, 83.75°E], the abrupt change in wildfire onset occurs following abrupt soil drying driven by rapid permafrost thaw (Figs. 1i, j, and 4c), and the timing of the abrupt wildfire onset is similar across the 50

ensemble members (Fig. 4c). After the abrupt wildfire onset over the region, the forced anthropogenic changes (ensemble means) of fire counts and the burned area gradually increase and their ensemble spread increases towards the end of 21st century (Fig. 4c). The ensemble means in the burned area over western Siberia reaches about 800 km² at the end of the 21st century (Fig. 4c) showing a dramatic intensification of the statistical moments of the fire probability distribution, which is reminiscent of an abrupt regime transition. Furthermore, the sudden increase in wildfires occurs primarily after sudden thaw-induced soil drying over ice-rich permafrost regions (Fig. S4). In contrast, there is no abrupt increase in wildfires in a warmer climate over historically fire-prone regions near the southern edge of the permafrost area, which can be explained by the absence of abrupt changes in soil ice and soil moisture (Fig. S5).

### Atmospheric and wildfire responses to soil moisture perturbations

To further isolate the impact of abrupt change in soil moisture content on Arctic and Subarctic wildfires, we conduct two additional idealized experiments with the CESM2 model, in which we reduce soil moisture in regions poleward of 40°N by 20% and by 40% (see Methods). We specifically focus on the summer season, which is the primary period for wildfires over the Arctic and Subarctic regions. In our idealized experiments, a decrease of 40% in soil moisture immediately leads to a substantial increase in surface air temperature of more than 5 °C across western Siberia and Canada in July (Fig. 5a, b), which can be attributed to a noticeable increase (decrease) in sensible (latent) heat flux from land to the atmosphere (Fig. S8). In conjunction with the anomalous surface warming, there is an anomalous decrease in relative humidity,

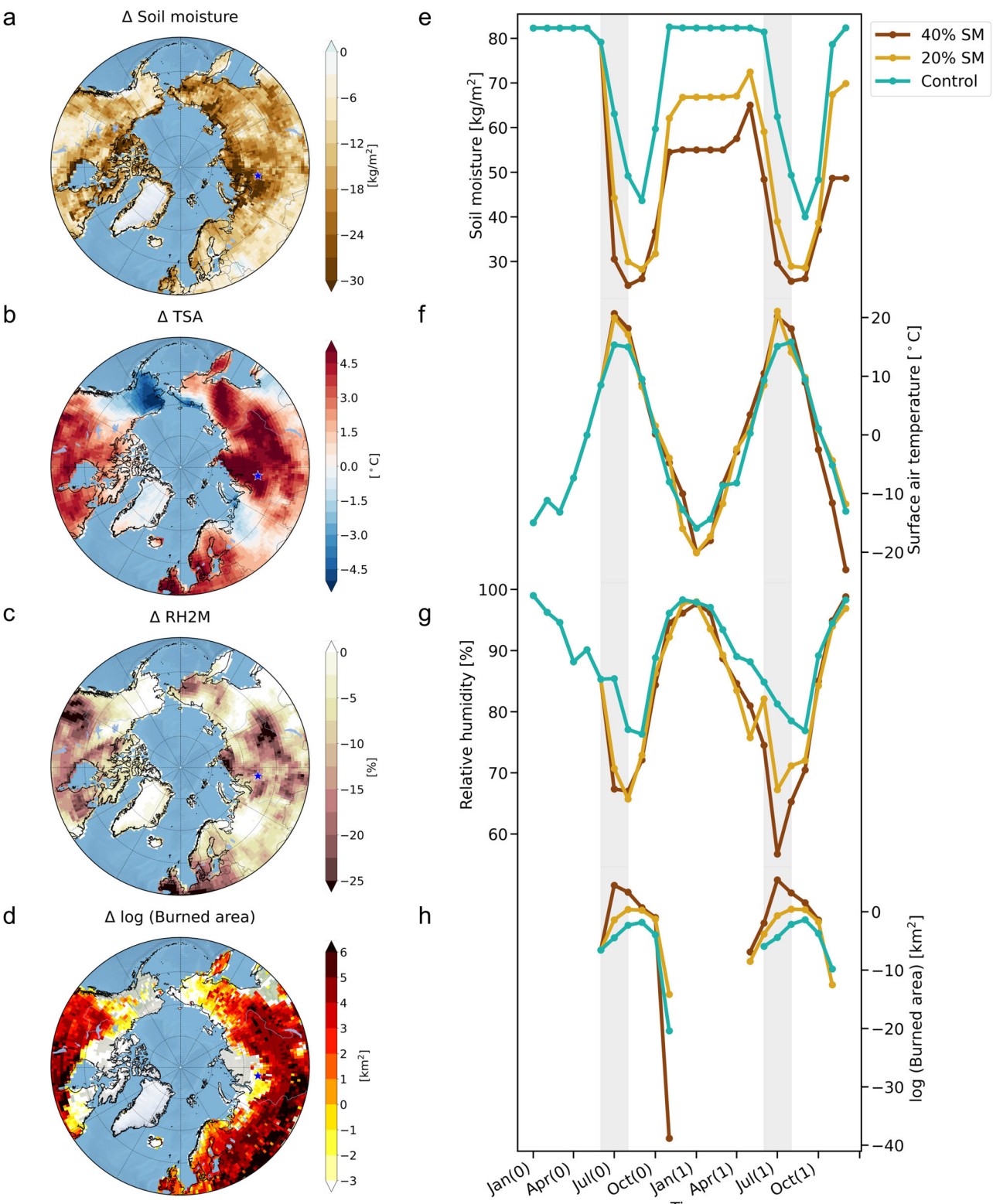

**Fig. 5 | Atmospheric and wildfire responses to soil moisture reduction in the idealized experiments using the CESM2.** The values represent differences between the response of a 40% soil moisture reduction perturbation experiment in July 2045 and a control simulation: (**a**) soil moisture in 0–10 cm depth (units: kg/m²), (**b**) surface air temperature (units: °C), (**c**) relative humidity at 2 m (units: %), and (**d**) logarithm of burned area [log (burned area)] (units: km²). Time evolution over Western Siberia (65.5°N, 83.75°E): (**e**) soil moisture over 0–10 cm depth (units: kg/m²), (**f**) surface air temperature (units: °C), (**g**) relative humidity at 2 m (units: %), and (**h**) logarithm of the burned area [log (burned area)] (units: km²) (blue: control simulation, yellow: 20% soil moisture reduction perturbation experiment, and brown: 40% soil moisture reduction perturbation experiment).

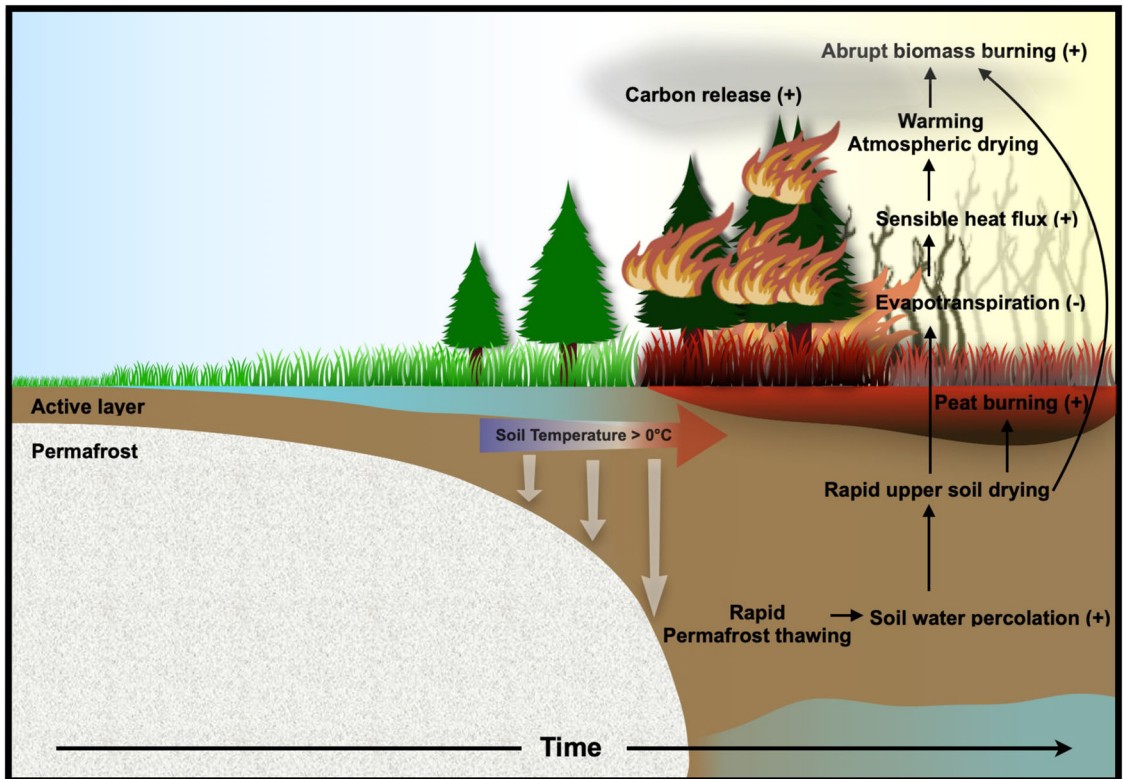

**Fig. 6 | Schematic diagram highlighting pathways for an abrupt increase in wildfires following permafrost thaw.** Permafrost thaw occurs in response to increasing greenhouse gas concentrations when soil temperatures exceed 0 °C. A rapid thaw over the ice-rich Arctic-Subarctic permafrost regions can trigger a subsequent abrupt drying of the upper soil due to increasing soil water percolation and an associated reduction in summer soil evaporation. This, in turn, increases sensible heat fluxes from the surface to the atmosphere, generating near-surface atmospheric warming and an increase in atmospheric dryness. These rapidly emerging conditions can promote wildfire. Moreover, positive trends in $CO_2$ fertilization in the CESM2-LE model further increase vegetation carbon stocks, which can serve as additional fuel for combustion, thereby contributing to the intensification of wildfires.

particularly in regions where surface temperature anomalies are higher (Fig. 5c). We also observe an anomalous cooling over Alaska (Fig. 5b), which may be influenced more by changes in the large-scale atmospheric circulation than by local land-atmospheric interactions. Comparing the 20% and 40% soil moisture reduction experiments reveals that these substantial changes in soil moisture and relative humidity lead to a nonlinear amplification of the burned area, as illustrated here for the western Siberian grid box (and others) in July in the simulation Year 0 (20% Exp.: 0.25 km², 40% Exp.: 92.5 km²) (Fig. 5d, g, h). Additionally, as a result of the slow recovery timescale, soil moisture in the perturbation experiments does not rebound to the pre-perturbation soil moisture state, at least within the first 2 years, which further prolongs wildfire activity (Fig. 5e, h). These findings are consistent with the results from our initial analysis using the CESM2-LE, confirming our original hypothesis that soil moisture plays a crucial role in Arctic-Subarctic wildfire activity.

## Discussion

In this work, our analyses of 50 ensemble simulations of the CESM2-LE under historical/SSP3-7.0 forcing demonstrate that permafrost thaw in the Arctic-Subarctic region can serve as a trigger for abrupt regime shifts in soil hydrological processes and regional wildfires (Fig. 6). In the model, rapid thaw in the ice-rich permafrost regions leads to an increase in soil water percolation due to increased permeability, which then causes a sudden upper soil drying. In addition, ground evaporation decreases in response to the abrupt shift towards a soil water deficit in summer, with an associated dramatic increase in sensible heat fluxes from the surface to the atmosphere. The abrupt increase in sensible heat fluxes can intensify the warming of near-surface air

temperature and enhance atmospheric aridity, further promoting wildfire intensity. The simulated abrupt increase in wildfires following rapid permafrost thaw is consistent with the findings of an earlier study[22], which analyzes the meteorological-based FWI. In contrast to the finding of the earlier study, we find that the CESM2-LE explicitly simulates the interactions between climate-vegetation-permafrost and fires, leading to a different representation of important coupled feedbacks and dynamics.

In the CESM2-LE simulations, subgrid-scale permafrost processes are parameterized in such a way that permafrost thaw in certain areas leads to a subsequent abrupt soil drying. This may be an over-simplification of the scale-dependent dynamics that can occur in geographically diverse permafrost regions. Polygonal permafrost landforms at the meter scale can significantly influence the hydrological cycle even at the watershed scale[12,15,19,32], but the representation of multi-scale interactions is one of the key challenges in ESMs. There have been new modeling developments to improve the representation of permafrost dynamics. For instance, experiments using a new parameterization with the Community Land Model 5 (CLM5) suggested that subsidence due to permafrost thaw can increase the surface water fraction[33]. In addition, small-scale simulations of the ice-rich lowlands have shown that thaw subsidence under waterlogged conditions can increase soil water saturation, thereby accelerating thaw[15], whereas under well-drained conditions soil water saturation decreases[15,32]. It is important to note that incorporating these new modeling developments may lead to results that differ from the hydrological responses to permafrost thaw in our study. Therefore, for future studies using ESMs, it will be essential that consider these recent modeling advancements and compare them with existing models.

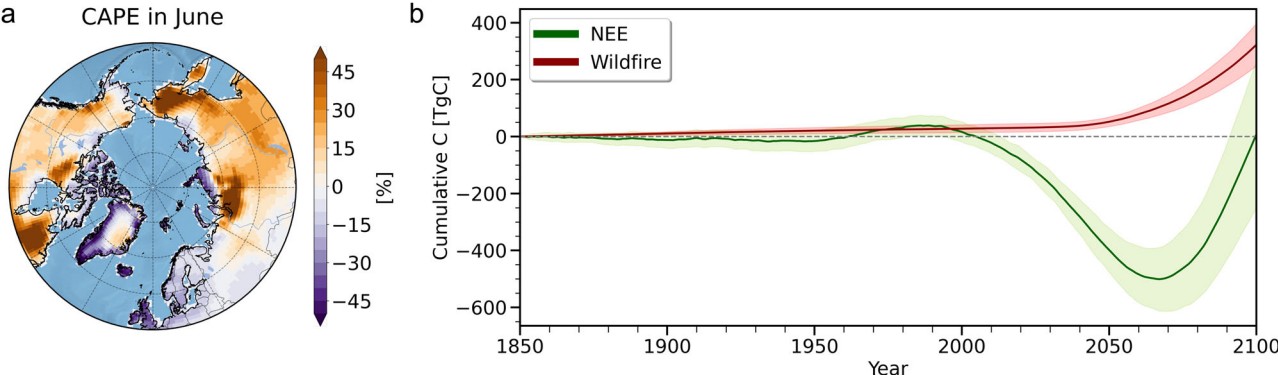

**Fig. 7 | Changes in convective available potential energy (CAPE) and carbon fluxes over the historical permafrost regions. a** CAPE in June for 2081–2100 relative to 1995–2014 (units: %) and (**b**) time evolution of cumulative Net Ecosystem Exchange (NEE), defined as the difference between net primary production and soil heterotrophic respiration (green), cumulative carbon emissions from wildfires (red) from CESM2-LE over the historical permafrost regions where the rapid changes in soil ice content and wildfires occur together (units: TgC). Bold red and green lines indicate ensemble means and shading indicates ±1 standard deviation of ensemble members.

To date, there is little direct observational evidence supporting that permafrost thaw-induced soil drying leads to an increase in wildfires on a large scale, as suggested by the CESM2-LE. However, observations have shown that permafrost thaw can lead to a drying of surface water bodies[34–38], atmospheric warming, increasing aridity, and decreasing soil moisture contribute to wildfire activity[1,5]. Other supporting evidence for some of the key processes highlighted in our study comes from studies on northern peat bogs, which show that drainage can increase carbon emissions due to peat burning[39,40] and that lower water tables under a warmer climate can potentially enhance the risks of peat burning[41].

Some other modeling caveats need to be mentioned: The CESM2-LE underestimates the observed burned area over the Arctic and Subarctic regions compared to tropical and temperate latitudes (Fig. S10). This could be due to the model's lack of explicit representation of changes in fire ignition. The model uses a fixed climatological lightning frequency for natural ignition without internal lightning noise. Additionally, future changes in lightning activity could be an additional important driver influencing Arctic wildfires[42]. Since lightning occurs mostly in convective systems with high values of convective available potential energy (CAPE), we can qualitatively assess, whether future warming is likely to increase lightning in the Arctic and Subarctic regions. The CESM2-LE simulates an increase of 50% in CAPE in June over western Siberia, far eastern Siberia, and Canada at the end of the 21st century (Fig. 7a). The CAPE has been used here as a proxy for lightning flash frequency[42,43], which serves as a natural fire ignition source. The substantial increase in CAPE-implied lightning in the CESM2-LE suggests that fire frequency may increase further over these regions towards the end of the 21st century, even beyond the levels simulated explicitly in the CESM2-LE due to permafrost thaw and soil drying.

The abrupt increase in wildfires over the historical permafrost regions can contribute to changes in net terrestrial carbon uptake. The quantitative estimation of carbon emissions due to the increase in wildfires is meaningful in the context of carbon trading and national greenhouse gas inventories. Our estimate from the CESM2-LE shows that wildfires occurring in permafrost regions experiencing abrupt changes would cumulatively release 322.6 ± 74.7 TgC towards the end of the 21st century and the cumulative net uptake by ecosystem production would reach about 8.9 ± 256.5 TgC in the same permafrost regions (Fig. 7b). Furthermore, the contribution of carbon release from wildfires to the net terrestrial carbon balance in these regions accelerates after the mid-21st century. However, the amount of carbon released by wildfires in the regions experiencing abrupt transitions represents only a relatively small contribution to the net terrestrial carbon fluxes occurring over the entire permafrost region (north of 50°N).

When representing the underlying processes in a large ensemble framework, the CESM2-LE presents an unprecedented resource for exploring and developing our understanding of feedbacks within the subarctic system and putting forced signals into the context of internally generated climate variability. However, as discussed above, it is also clear that there are additional processes that can contribute to modulating the permafrost thaw-related wildfire activity. These include (i) representation of ground subsidence that can occur in response to excessive ice melting, (ii) consideration of incoming lateral flows from adjacent landscapes into local grid cells in soil hydrological cycles (iii) enabling climate interactions of $CO_2$ and aerosol emissions from fire through ESMs projection with emissions pathways, (iv) inclusion of parameterizations of lightning that evolve with the climate state, (v) allowing for more realistic consideration of potential vegetation transitions through the vegetation demographic processes, and (vi) reflecting effects of fire thermal properties on permafrost thaw. Such processes need to be included in future generations of ESMs to obtain more reliable projections of climate/carbon cycle interactions in high latitudes.

## Methods
### Observations and reanalysis datasets
We used the burned area and biomass burning carbon emissions from the monthly Global Fire Emissions Database version (GFEDv4)[44] to evaluate simulated wildfire activity with observations over 1997–2021. We used reanalysis datasets on ALT and ground temperature over the Northern Hemisphere from the European Space Agency (ESA) Climate Change Initiative permafrost (CCI-PF) project for the period 1997–2019[45,46], with this being based on MODIS Land Surface temperature merged with downscaled the European Centre for Medium-Range Weather Forecasts (ECMWF) Reanalysis v5 (ERA5) reanalysis near-surface air temperature data. We used observed ALT from the Circumpolar Active Layer Monitoring (CALM) Program Network during the period 1997–2014[47]. 2 m air temperature from the ERA5 reanalysis was used during the period 1997–2019[48].

### CESM2 large ensemble simulations
We used data from the ICCP/NCAR CESM2-LE project[23], a single model large ensemble that was initialized in 1850 with different initial conditions based on micro and macro perturbations[49]. The large ensemble uses CMIP6 forcings, consisting of historical (1850–2014) and SSP3-7.0

(2015–2100) pathways (i.e., high greenhouse gas emissions scenario). Here, we chose 50 members, for which consistency in biomass burning was obtained through smoothing over 1990–2014 relative to the CMIP6 protocols[20].

The CESM2-LE uses the CLM5, the most recent of the CLM family of land models in the CESM[50]. The CLM5 incorporates comprehensive permafrost-related soil thermal and hydrological dynamics, carbon cycle dynamics, and process-based fire parameterization. The fire parameterization in the CLM5 encompasses processes for fire occurrence, fire spread, and fire impact (fire carbon emissions)[51–53] and updated the dependence of fire occurrence and spread on fuel wetness[50]. It also represents agricultural fires, deforestation fires, peat fires, and non-peat fires outside of cropland and tropical forests respectively. The fire activity in the CESM2-LE is mainly governed by weather and climate conditions (e.g., relative humidity, soil moisture, surface air temperature, and wind speed), climatological lightning and human-caused ignition, fuel load, and vegetation type. In addition, it is also important to note that the wildfire-induced carbon release does not affect the atmospheric radiation in our simulations, since the CESM2-LE was carried out as a concentration-driven experiment.

The CESM2-LE represents reasonably well the overall spatial pattern of ALT (correlation coefficient between the CESM2-LE and CCI-PF: 0.53, $p < 0.00001$). However, the CESM2-LE does not capture the observed ALT over Alaska and quantitatively underestimates the magnitude of ALT over the permafrost regions (Fig. S9). Furthermore, the CESM2-LE captures the global spatial patterns of observed burned area (correlation coefficient: 0.54, $p < 0.00001$) (Fig. S10a, c). However, it is important to note that the model simulations underestimate the observed burned area in higher latitudes (north of 60°N) (GFED4.1 s: 13,081 $km^2$, CESM2-LE: 2036 $km^2$) (Fig. S10b, d). Particularly, the CESM2-LE does not well represent the observed burned area in Alaska and northwestern Canada (Fig. S10b, d).

### Idealized soil moisture reduction experiments

In the two idealized experiments, we impose artificial reductions in soil moisture content within all soil layers by 20% and 40% over the high latitudes (north of 40°N) from the July 1st, 2045 model state (pre-thaw condition) in one ensemble member of the CESM2-LE. The choice of 20% and 40% reduction in soil moisture for the idealized experiments is made to account for the abrupt 20–40% reduction in soil moisture following permafrost thaw in the CESM2-LE. Subsequently to this one-time perturbation on July 1st, 2045, which resets only the initial conditions in the CLM5, we allow soil moisture to evolve freely over time within the experiments under the same forcings as the CESM2-LE simulations. The experiments are run for 2 years and compared to the corresponding CESM2-LE ensemble member, which is treated as a control simulation.

### Detection of abrupt change over the historical permafrost regions

To identify abrupt changes over the permafrost regions, we conducted a change point analysis based on a linear regression model[24,25]. The detection algorithm considers coefficient shifts based on the Bayesian Information Criterion and Residual Sum of Squares. To change in each region during 1850–2100, we considered the case where the largest abrupt change could occur in each area. In particular, we defined the rapid changes in ALT and subsurface runoff as a more than twofold increase in ALT and subsurface runoff over 20 years. Grid points exhibiting a more gradual transition in ALT of more than 20 years were excluded. Similarly, we also defined the rapid changes in soil ice as a more than 30% decrease in soil ice and the rapid changes in soil moisture as a more than 20% decrease over 20 years. The rapid changes in burned area were defined as a more than twofold increase in burned area over 20 years in the regions experiencing abrupt

changes in soil ice. We then applied change point detection to each ensemble member and calculated the median value from the 50 ensemble members to identify robust characteristics in the ensemble members. We analyzed abrupt changes in ALT, soil ice content, soil moisture, and burned area over the period 1850–2100 using the detection algorithm.

## Data availability

CESM2-LE model output is available at: https://www.cesm.ucar.edu/projects/community-projects/LENS2/data-sets.html. The data files used for the main figures are available at: https://doi.org/10.5281/zenodo.11239502 GFEDv4: https://daac.ornl.gov/cgi-bin/dsviewer.pl?ds_id=1293. ESA CCI: https://climate.esa.int/en/odp/#/project/permafrost. ERA5: https://cds.climate.copernicus.eu/#!/search?text=ERA5&type=dataset.

## Code availability

The R package for structural change is available at: https://cran.r-project.org/web/packages/strucchange/index.html. The codes supporting the findings of this study are available from the corresponding authors upon reasonable request.

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

## Acknowledgements

I.-W.K., A. T., J.-E.K., K.B.R., and S.-S.L. were supported by the Institute for Basic Science (IBS) IBS-R028-D1. K.B.R. was supported by the World Premier International Research Center Initiative of the Ministry of Education, Culture, Sports, Science and Technology of Japan. H. L. was supported by the Research Council of Norway project 328922. W.R.W. was supported by the National Science Foundation (NSF) award number 2031238. The simulations were conducted on the IBS/ICCP supercomputer "Aleph," 1.43 Petaflops high-performance Cray XC50-LC Skylake computing system with 18,720 processor cores, 9.59 PB storage, and 43 PB tape archive space. We also acknowledge the support of KREONET.

## Author contributions

I.-W.K., A. T., and H. L. developed the scientific framing of the manuscript. I.-W.K. performed the main analysis. J.-E.K. conducted the idealized experiments. K.B.R. and S.-S.L. contributed to the CESM2-LE project. I.-W.K., A. T., H. L., J.-E.K., K.B.R., S.-S.L., and W.R.W. discussed the results and contributed to the writing of the manuscript.

## Competing interests

The authors declare no competing interests.
