## [Peer Review File · Nature Communications]

Abrupt increase in Arctic-Subarctic wildfires caused by future permafrost thawREVIEWER COMMENTS

Reviewer #1 (Remarks to the Author):

The study presents findings about future projections of wildfire activity in the Arctic and Subarctic permafrost region. The authors predicted an increase in burned area potentially caused by future permafrost thawing and the subsequent changes in hydrology. Although the findings have a significant impact and are of general scientific interest, I do not see the novelty of this work compared to a previous study (Teufel et al, 2019), which also predicted a change in fire regime after abrupt permafrost thaw and subsequent decrease in soil moisture using a climate model. That study raised critiques regarding the validity of the abrupt changes in permafrost using simulation (O'Neill et al., 2020). Also, I have major concerns about the lack of empirical evidence that supports the conclusions presented by Kim et al. Additional evidence is required to support the link between permafrost thawing and increased fire activity. Lastly, I have concerns regarding the lack of engagement with the existing literature in the Discussion section.

The authors used a model that predicted an increase in burned area increases after future permafrost thawing. However, the association between permafrost thawing and wildfire activity needs to be validated with empirical evidence. Existing satellite products provide observations that can demonstrate whether past acceleration in fire activity were indeed preceded by a deepening of the active layer and a decrease in soil moisture in the arctic and subarctic. This evidence is critical to achieve the aim of the study and support the projections of the model. Although the authors provided a comparison between modelled and observed trends, the comparison do not depict a potential association between permafrost thawing and burned area.

Another concern relates to the projected abrupt increase in evapotranspiration (fig. S7). The authors attribute the abrupt increase in evapotranspiration "due to an increase in photosynthetic carbon uptake, which is determined by both CO₂ fertilization and warming effects". However, both warming and CO₂ fertilisation are unlikely to change abruptly in the 21st century and, thus, the change in evapotranspiration is equally unlikely. Moreover, plant evapotranspiration should decrease as a result of low soil moisture. It is important to understand the reason behind the abrupt increase in evapotranspiration because this variable is linked with fuel moisture and, hence, fire risk.

The Discussion section falls short of engaging with the existing literature. The Discussion section only includes three references, none of which were used to contrast the findings of the study. By citing previous studies, the authors have an excellent opportunity to further support the model's projections. For instance, 1) it is known that soil moisture, more concretely lowering the water table, can increase the extent of peat fires (Turetsky et al., 2015). 2) Other studies, included in the Introduction section, explain the changes in hydrology, supporting the model's outputs. 3) The model projections could be compared with future projections in burned areas obtained in previous studies.

Two paragraphs were placed in the incorrect section of the manuscript. First, the last paragraph of the Introduction (L62-L70) presents results about model validation. This sentence belongs to the Results section rather than the Introduction. Also, please note that, according to the formatting guideline of the journal, "The final paragraph must begin with a phrase like "In this work" or "Here, we show", and contain a summary of the major results and conclusions of the current work, written in the present tense.". The Discussion section also presents additional results in L221-224 and L228-236. This paragraph also belongs to the Results section.

Specific comments:

L15: The title and abstract only mention the subarctic region. However, results and figures also include the Arctic region.

L17-18: The causal relation of permafrost thawing on increased wildfire activity has not been demonstrated. The authors have pointed this out in L43, as part of the justification of the study. Thus, the statement L17-18 requires rephrasing.

L26: Instead of mentioning 'several orders', specify the order or range of magnitude of the abrupt intensification.

L29: 'wet environment' might mislead the reader. Although evapotranspiration is low, the tundra biome receives low levels of precipitation and, thus, could be also labelled as a dry environment.

L35-39: 'low soil moisture' suits better than 'arid soils' in this sentence. Also, I suggest replacing 'soil wetness' by 'soil moisture' to keep a consistent terminology throughout the manuscript.

L35: Please check whether Ref. #6 actually supports the statement 'arid soils increase the probability, frequency, and spatial extent of fires.'

L53 mentions 'previous studies' but only one reference is provided.

L78-80: The maps in Fig. S3a,b does not show the association between warming and ALT in a clear manner. In fact, although the southern margin of the permafrost region presents a pronounced trend in ALT, this region presents a warming trend that is lower than the northern part of the permafrost region. I suggest a scatter plot between warming trends and ALT trends to elucidate the association.

L89: Does 'anthropogenic effects' refer to 'anthropogenic warming'?

L115-116: Could rapid permafrost thawing increase soil moisture in flat terrains and depressions? If that is the case, the increase in fire risk could be highly spatially dependent. I wonder if the model can make predictions based on geomorphology.

L118-122: The justification for focusing on western Siberia is not sufficient. The same justification could be used for selecting another region in the arctic or subarctic. At the very least, these analyses should also focus on a fire-prone region (for instance, Sakha republic) and compare the trends in both regions. Ideally, the study should subdivide the arctic and subarctic in subregions and analyse the trends in each subregion to provide a comprehensive overview of future projections.

L122: The authors must clarify the month that was selected for these results. According to the caption in Fig. 1, these results were obtained for October. Also, the manuscript must explain why October was selected for this analysis while July was selected in other figures.

L134-139: These results might be substantially different in other regions of the arctic. Consider including more subregions for this analysis.

L148-153: Following previous comment, Fig. 3 and Fig. 4 show that the change in fire regime will only occur in certain regions on the arctic and subarctic. The manuscript should clarify that the projected abrupt soil drying and subsequent increase in fire activity will occur only in these specific regions. Moreover, according to Fig. 3a, projected evapotranspiration will increase in historical fire prone regions of the subarctic. Given the direct link between evapotranspiration and fuel moisture, these results might be interpreted as vegetation will be less susceptible to fire.

L161-162: The terms 'abrupt' and 'rapid' increase/decrease are used subjectively. Surface temperature also shows a significant change comparable to ground heat.

L172: I suggest providing changes in burned area in absolute terms (km²) or density of burned area (km² burned per pixel size) instead of percentages. The >1000% might mislead the reader. This extreme change might be only possible because western Siberia and eastern Canada present no current fire activity.

L177-179: Following previous comment, the changes in the burned area in absolute terms seem unrealistic. According to Fig. 4c, the burned area will increase up to 1000 km² only in a pixel in Western Siberia by the end of the century. Information on the pixel size is key to understanding this figure panel.

O'Neill, H. B., et al. "Permafrost thaw and northern development." *Nature Climate Change* 10.8 (2020): 722-723.

Teufel, Bernardo, and Laxmi Sushama. "Abrupt changes across the Arctic permafrost region endanger northern development." *Nature Climate Change* 9.11 (2019): 858-862.

Turetsky, Merritt R., et al. "Global vulnerability of peatlands to fire and carbon loss." *Nature Geoscience* 8.1 (2015): 11-14.

Reviewer #2 (Remarks to the Author):

Summary

The authors used a large ensemble simulation performed by the state-of-the-art Earth system model which considers the coupling between permafrost and hydrology. They investigated the relationship between 1) permafrost thawing, 2) thawed water percolation to a deeper layer, 3) soil drying at the upper layer, 4) an increase in surface air temperature and decrease in relative humidity, and 5) an increase in wildfire occurrences. I think this topic is very important and interesting, but I believe the conclusions are not supported by strong evidence. Therefore, I should say that the current manuscript does not satisfy the criteria for the publication of Nature journals. The reasons are explained as follows. I recommend that the authors consider these points and improve the manuscript.

General Comments

1. First, the causal relationships from the above 1) to 5) are shown mainly based on a "representative" grid point in northwestern Siberia (65.5N, 83.75E). Therefore, I couldn't really understand what was happening elsewhere. The authors show the maps of changes in active layer thickness, soil ice, soil moisture, runoff, evapotranspiration, Bowen ration, and burned area by performing the "change point analysis" (Figure 1-4). I am not sure that the changes in these variables are really caused by permafrost thawing, because the map of ALT changes (Figure 1a) is not consistent with other variables (Figure 1c, 1e, 2b, 3b, 3d), and similarities and differences between these distributions are not discussed in the manuscript. In the first place, will the permafrost thaw rapidly (abruptly) at any location? If the changes in permafrost and other variables are gradual and not abrupt, what do the results of "change point analysis" mean? This analysis assumes that "one abrupt change occurs over each region" (Line 295). In fact, the distribution in Figure 1a (the timing of ALT change) appears implausible. The timing of changes in the Arctic Ocean coastline at higher latitudes is earlier than in inland areas at lower latitudes. Furthermore, the distribution of ALT appears to be uniform. The authors should explain the changes in permafrost thawing and other variables

should be abrupt at any location (and what determines the time scale of thawing) because they assume “one abrupt change occurs over each region” in their analysis. In addition, the readers would be interested in the time sequence at locations other than the representative grid cell. The authors should show the time sequence with regional average for different regions.

2. I could understand the causal relationship from 1) to 4) at the representative grid cell (65.5N, 83.75E). However, the relationship between the permafrost thawing and 5) increase in wildfire occurrence is not shown with strong evidence. The author claims that wildfires increase from 2050 (Figure 4c), but it is unclear whether this is due to the permafrost thawing or changes in atmospheric conditions. In addition, the wildfire occurrence increases as shown in Figure 7b, but the role of permafrost thawing in Figure 7b is not clear (although it is the result of permafrost domain, changes in atmospheric conditions could increase the wildfire over there). The wildfire increases in the sensitivity experiments (Figure 5), but the experimental setting, i.e., abrupt 40% (80%) reductions within all layers over the high latitudes (> 40 N), should not correspond to the conditions after permafrost thawing (the authors should explain why they chose these experimental settings).

3. The authors discuss the future projection of permafrost and hydrology based on their simulation results, but they should consider other possible processes not described in CESM2. In the literature of future permafrost projections, the increase in methane emissions by wetland expansions is discussed. Are there any areas where the wetlands are expanding in CESM2? Is the “abrupt drying” after permafrost thawing in CESM caused by the model not considering the more complicated phenomena (e.g., thermokarst formulation)? I understand it is not easy to consider the complicated phenomena in the global model, but I hope the possibility it should be discussed in the paper.

Individual comments

Line 24: As I commented above, the authors show the results based on a single grid point. I am not sure if these results show the regime shifts over the subarctic.

Line 62: This paragraph is the results of model simulations, and thus it is not suitable for the introduction.

Line 100: “Historical permafrost regions (> 70 N)”. Do you refer to “permafrost regions” as the entire region over 70 N? The phrase “permafrost regions” is used in the manuscript, but they are not consistent (for example, > 55 N in Figure 7). Why you change the region? The term “permafrost” is usually defined as the region where the ground temperature is below 0 degC for at least two years. The authors should define this term and explain why they chose different regions. The definition of “historical permafrost domain” is (probably, only) shown in the caption of Supplementary Figure 4 (Line 40 in SI) as “the area where ALT is less than 3m”, but why the authors choose this threshold (ALT < 3 m)?

Line 105: “abrupt increase in ALT”. As shown in Supplementary Figure 3f, the trend in ALT is about 0.1 m/23yr. On the other hand, the increase in ALT shown in Figure 1h is about 40 m during about 20 years. Are these results in Supplementary Figure 3f and Figure 1h consistent? Is the ALT increase shown in Figure 1h physically reasonable (what determines the time scale of permafrost thawing)?

Line 107: “The timing of the 0 degC 10 cm ground temperature exceedance serves as a good proxy for the abrupt responses in soil ice and soil water”, but why is the timing of “10 cm ground temperature exceeds 0 degC” so important?

Line 128: If you can show the results of “vertical hydraulic conductivity”, it is helpful to support the

hypothesis. I am interested in how they change during the simulation.

Line 134: "Transpiration and evaporation from the plants increase ... due to increase in photosynthetic carbon uptake, ...". This is interesting. I think the CO₂ fertilization effect reduces transpiration by the changes in stomata. Can you add more explanation?

Line 163: "This can intensify surface warming, thus leading to a rapid decline in relative humidity ... despite a smaller change in the actual amount of water vapour". This explanation is difficult to follow. Is the decline in relative humidity simply caused by 1) surface warming due to less latent heat and 2) less water vapour due to less evaporation?

Line 185: As I described above, why did the authors choose the abrupt decrease by 40 or 80 % in regions poleward of 40 N? The reason why you chose these settings should be explained. Otherwise, we cannot understand the meaning (how it is related to the realistic future projection) of the idealized experiments. To show the impact of permafrost thawing, the experiments starting from the "post-thaw" conditions can be useful to show the causal relationship between permafrost thawing and the increase in wildfire occurrence, for example.

Line 234: "Cumulative net uptake by ecosystem production (NEP) over the same permafrost regions (2.03 ± 0.51 PgC)". It seems small compared to the annual land carbon uptake is reported about 3 GtC per year (<https://essd.copernicus.org/articles/14/4811/2022/>). Is this number, correct?

Line 243: As commented above, the thermokarst formation and wetland expansions would be also an important factor in projecting the future permafrost and hydrological change.

Figure 1: In some regions along the arctic coast, abrupt change in ALT is earlier than the lower latitudes. I expect that the time of abrupt change in the higher latitudes become later than the lower latitudes.

Figure 3: Only this figure shows the difference between pre- and post-thaw. This seems reasonable to show the impacts of permafrost thawing. Other maps (Figures 1, 2, and 4) are calculated as the pre- and post-abruptness: as I commented above, the results are difficult to understand.

Figure 7b: Since this is cumulative carbon emission, the values of "Fire" and "GFED4.1s" should be positive (negative means decrease?). I also noticed a large difference between the model simulations and GFED. This should be discussed in the manuscript (even though the difference between the simulations and observation averaged over 1997-2004 are discussed in Line 66).

Reviewer #3 (Remarks to the Author):

The authors present a comprehensive analysis of some of the effects of future permafrost thawing on land surface processes, with a particular focus on the interactions between permafrost, soil hydrology and wildfire. Their analysis is based on an ensemble of 50 simulations performed using a state-of-the-art Earth system model, which includes permafrost-related soil thermal and hydrological dynamics, carbon cycle dynamics, and a process-based fire parameterization. The manuscript is well organized, easy to follow, and effectively uses high-quality graphics to illustrate the main points of the analysis.

The manuscript is consistent with previous research which showed that permafrost thawing can lead to surface soil drying, and that this drying affects upward and downward fluxes of heat and moisture. As acknowledged in the manuscript, the conclusion that permafrost thawing-led soil drying has the

potential to increase wildfire activity was first presented in Teufel and Sushama, 2019 (hereafter, TS19). It is important to note that the model used for the current study incorporates carbon cycle dynamics, including a process-based fire parameterization, which was not the case for TS19. Including the carbon cycle enables the analysis to account for future changes in vegetation, and the fire parameterization allows for the explicit quantification of fire counts, burned area and fire type (e.g., biomass vs peat). In addition, the ensemble of 50 members in the current study (10 times larger than in TS19) increases the statistical robustness of the conclusions.

Permafrost landscapes are complex, and some of this complexity cannot be captured at Earth system model scale (~100 km horizontal resolution). For example, subgrid orography will have a strong influence on moisture pathways as permafrost thaws, which implies that some parts of a gridcell might experience wetter soil conditions, while others dry out. This complexity does not invalidate the conclusions of the study, as many regions are projected to experience drying and the ensuing consequences detailed in the manuscript. However, it would be valuable for this complexity to be discussed in the manuscript and to acknowledge that the response at local scales might be significantly different from the CESM2 projections.

Other comments:

Title can lead to confusion. The current title: "Abrupt increase in subarctic wildfires amplified by future permafrost thawing" can be interpreted to imply that an abrupt increase in wildfires would occur in all cases, and that permafrost thawing only serves to amplify said abrupt increase. The abstract and manuscript make it clear that the abrupt increase in wildfires is actually a consequence of permafrost thawing, clarity that should also be reflected in the title.

L18: "which can also contribute" should be "which also contribute".

Reference 18 is missing title and journal.

Supp. Fig. 1. Caption mentions unit as: "(unit: m/yr)". This should either be removed or changed to the correct unit shown on the panels (i.e., m).

L63-64. The origin of the correlation coefficient equal to 0.53 is unclear. Is it between CESM2 and CALM? Or between CESM2 and CCI-PF?

L64-65. The discrepancies between CESM2 and observations over (western?) Siberia are mentioned. However, the much larger discrepancies over the majority of Alaska and northwestern Canada are not mentioned, where CESM2 fails to capture large areas of discontinuous and even continuous permafrost.

L67-70. A quantification of the underestimation of the observed burned area over the (sub)arctic would be very valuable (e.g., observed vs. modelled burned area north of the 60th parallel). It also seems odd that Alaska has zero burned area in CESM2, while comparable areas of Canada and eastern Siberia do experience some fire.

Supp. Fig. 2. Panels (b) and (d) show areas north of 55 degN. All other figures with similar panels show areas north of 50 degN.

Supp. Fig. 3. The meaning of stippling for panels (a), (b) and (c) should be mentioned in the caption.

L75. Is the word "air" correct here?

L76. If ERA5 is the source of surface temperature, it should be mentioned here in addition to CCI-PF

L84-85. The message is understandable, but the terminology used here is not correct. E.g., "outside the range" would only be appropriate if the observed value was higher than the maximum value in CESM2-LE (or lower than the minimum), which is not the case. Also, "projected changes" is usually reserved for future climate, while here the analysis focuses on the historical period.

L103. Should be "(Fig.S4)" instead of "(Fig.S3)".

Units of "soil moisture @ 10cm" and "soil ice @ 3m". The used units of kg/m² only make sense when these quantities are integrated over a layer of certain depth (e.g., soil moisture from 0cm to 10cm). For values at a certain depth, the correct units are kg/m³ or m³/m³. Please correct throughout the manuscript.

Supp. Fig. 4. There are two panels (e), and both have wrong units (division by time is missing).

Inconsistent definition of historical permafrost domain. For Fig. S4 (> 55 °N, ALT is less than 3m for 1850-1869). For Fig. S5 (55-60°N 60-120°W, 63-70°N 40-90°E, 60-70°N 170-180°E).

Fig. 2(b). Panel has incorrect units (should be mm/month).

Fig. 2(e). Why anomalies relative to 1850-2100? Given that the 1850-1950 period looks stable, why not take anomalies relative to that period?

What is the value of soil depth at 65.5°N, 83.75°E ? Fig. 1(h) indicates it as being close to 4m. However, Fig. 2(d-e) shows that there is moisture below 4m.

Supp. Fig. 5. (a) Label of y-axis should be "Soil moisture @ 10 cm [Year]".

Supp. Fig. 7. Units are specified as mm/yr in the caption, but the plotted values would make much more sense if the units are mm/month. Use "snowfall" instead of "snow" to avoid ambiguity. What is the difference between "(b) surface water storage runoff" and "(c) surface runoff"? Also, difference between "(f) transpiration" and "(g) plant evaporation"?

L122-123. Typically, relative change is expressed as a fraction of the reference value, so it would be: $(51.4 - 71.1)/71.1 = -28\%$ (a 28% decrease).

L138. Use "sparse vegetation", because "reduced vegetation" suggests that vegetation is decreasing, which is the opposite of what is happening in CESM2 (Fig. S7f-g).

L158. Is it 19.2 instead of 12.9 ?

L162-164. The sensible heat flux actually acts to cool the surface, as it transfers heat from the surface to the atmosphere. Thus, the warmer surface is the one responsible for the increased sensible heat flux, not the other way around.

Fig. 5(b). Please comment on the mechanisms behind the strong cooling observed over Alaska.

L220-221. A word seems to be missing between "to" and "lightning".

L228-236. Please include a comment on the black line in Fig.7b (GFED4.1s).

L261. Either "surface temperature" or "air temperature at 2m".

L266. Missing word between "1850" and "different".

Is there a dependency between surface soil wetness and soil albedo in CLM?

Reply to three reviewers' comments

We thank the reviewers for their constructive comments and valuable suggestions aimed at enhancing the quality of the manuscript. To address the relevant points raised by the reviewers, we have conducted additional analyses and present a comprehensive point-by-point response below. The original comments are marked in black, and our responses are highlighted as blue text.

REVIEWER COMMENTS

Reviewer #1 (Remarks to the Author):

The study presents findings about future projections of wildfire activity in the Arctic and Subarctic permafrost region. The authors predicted an increase in burned area potentially caused by future permafrost thawing and the subsequent changes in hydrology. Although the findings have a significant impact and are of general scientific interest, I do not see the novelty of this work compared to a previous study (Teufel et al, 2019), which also predicted a change in fire regime after abrupt permafrost thaw and subsequent decrease in soil moisture using a climate model. That study raised critiques regarding the validity of the abrupt changes in permafrost using simulation (O'Neill et al., 2020). Also, I have major concerns about the lack of empirical evidence that supports the conclusions presented by Kim et al. Additional evidence is required to support the link between permafrost thawing and increased fire activity. Lastly, I have concerns regarding the lack of engagement with the existing literature in the Discussion section.

The authors used a model that predicted an increase in burned area increases after future permafrost thawing. However, the association between permafrost thawing and wildfire activity needs to be validated with empirical evidence. Existing satellite products provide observations that can demonstrate whether past acceleration in fire activity were indeed preceded by a deepening of the active layer and a decrease in soil moisture in the arctic and subarctic. This evidence is critical to achieve the aim of the study and support the projections of the model. Although the authors provided a comparison between modelled and observed trends, the comparison do not depict a potential association between permafrost thawing and burned area.

We thank the reviewer #1's thoughtful evaluation of the manuscript. We have revised the manuscript in response to the reviewer's comments.

Teufel, B. et al., 2019¹ used the limited area version of the Global Environmental Multiscale (GEM) model with the land surface scheme CLASS v.3.6. They used a fire weather index (FWI) to explain the abrupt change in wildfires. The FWI considers solely the atmospheric conditions such as temperature, relative humidity, precipitation, and wind without explicitly incorporating soil moisture (fuel moisture) and fuel condition (fuel amount and fuel types).

In contrast, the CESM2 is a fully-coupled model which includes the atmosphere, permafrost, hydrology, carbon cycle (including vegetation dynamics), and actual fire dynamics. Therefore, the CESM2-Large Ensemble (CESM2-LE) explicitly computes the fuel (vegetation and peat) of wildfires and the associated carbon release into the atmosphere. It also simulates how vegetation affects the hydrological and surface energy cycles over the Arctic and Subarctic regions in response to greenhouse warming, providing valuable insights into the underlying physical mechanisms that drive wildfire activity.

In Teufel, B. et al., 2019, in-depth analysis of permafrost thawing and its contribution to abrupt changes in land and atmospheric processes in a warmer climate was lacking. The previous study focused only on when permafrost first stops acting as a hydraulic barrier, while our investigation examines the differences in soil moisture between the regions experiencing the rapid and gradual permafrost thawing (see Figs. S6-7). We additionally conduct a series of idealized soil moisture reduction experiments to further elucidate the causality between abrupt soil drying and Arctic wildfires in the transient climate simulations.

O'Neill, H. et al., 2020² pointed out that there is no historical evidence that shows a rapid transition of the soil environment in response to permafrost thawing². However, other studies presented clear evidence for a rapid soil water drainage following permafrost thawing³⁻⁵ (Fortier, D. et al., 2007, Arp, C. D et al., 2023, and Swanson, D. K. et al., 2019) and rapid soil drying and marked vegetation shift after ice wedge degradation⁶ (Perreault, N. et al., 2016).

Furthermore, O'Neill, H. et al., 2020 raised some concerns regarding the Canadian Land Surface Scheme (CLASS)'s assumptions⁷ on permafrost hydrology and its simplistic representation of processes. It is evident that any parameterization of permafrost/soil hydrological coupling in global climate models with 100 km grid resolution would have to simplify the complex small-scale polygon-type dynamics of drainage processes. Certainly, a scale-gap will remain in earth system models for many more years, until they will be able to resolve sub-kilometer scale atmosphere, permafrost and soil dynamics. Nevertheless, our global-scale CESM-LE simulations can still provide important insights into atmosphere-fire-permafrost-hydrology coupling and potential regime changes induced by permafrost thawing that can neither be resolved with small-scale uncoupled permafrost process models, nor with offline calculations of Fire-weather indices. We therefore think that our simulations provide an important contribution towards understanding the complex interplay and coupling between warming-induced permafrost thawing, soil drying and rapid increases in fire activity, that could not be resolved previously.

We have also updated the references in the discussion section and engage now in a broader discussion. Our revised text now includes a more detailed discussion of the comments brought up by the reviewer (see blue revised text in main manuscript).

Lines 231-235: The simulated abrupt increase in wildfires following rapid permafrost thawing is consistent with the findings of an earlier study, which analyzes the atmospheric Fire Weather Index. In contrast to this study, our CESM2-LE simulates the interactions between climate-vegetation-permafrost and fires explicitly which leads to a different representation of important coupled feedbacks and dynamics.

The reviewer has raised the concerns about the lack of empirical evidence. As noted by the reviewer, there is currently little direct observed evidence indicating that permafrost thawing-induced soil drying leads to an increase in wildfires. Although it is difficult to establish a direct link between observed permafrost thawing and wildfires, the risk of permafrost thaw and Arctic wildfires has noticeably increased. For example, Figure R1 shows that there has been a significant increase in carbon emissions from biomass burning with a decrease in soil moisture in certain Arctic regions (63-90°N, 120-170°E) over the period of 2018-2021. The observed increase in Arctic wildfires is associated with drier atmospheric conditions and ground cover resulting from a significant increase in surface air temperature^{8,9} (Descals, A. et al., 2022, Kharuk, V. I. et al., 2022). In addition, the active layer in the Arctic regions (63-90°N, 120-170°E) has gradually deepened over the period 2002-2021 (Fig. R1).

Previous studies have presented empirical evidence establishing potential connections between permafrost thawing and wildfires. Recent studies have documented long-term trends in landscape and large-scale drying over the permafrost zone^{10,11} (Webb, E. E. et al., 2022 and Webb, E. E. et al., 2023). This landscape-scale drying is mainly attributed to lake drainage rather than changes in precipitation and evapotranspiration¹⁰ (Webb, E. E. et al., 2023). Additionally, other studies suggested that drainage in a northern peatland can cause carbon losses from peat burning^{12,13} (McCarter, C. et al., 2021 and Turetsky, M. et al., 2011). Therefore, we believe that this observed evidence supports a potential link between permafrost thawing and wildfire under stronger global warming.

Our revised text now includes a discussion of the comments brought up by the reviewer (see blue revised text in main manuscript).

Lines 252-259: To date there is little direct observational evidence supporting that permafrost thaw-induced soil drying leads to an increase in wildfires on a large scale, as suggested by the CESM2-LE. However, observations show that permafrost thawing can lead to drying of surface water bodies and that atmospheric warming and increasing aridity, and decreasing ground moisture contribute to wildfire activity. Other supporting evidence for some of the key processes highlighted in our study comes from studies on northern peat bogs, which show that drainage can increase carbon emissions due to peat burning and that lower water tables under a warmer climate can potentially enhance the risks of peat burning.

Figure R1 Time evolution of active layer thickness (ALT), near-surface soil moisture, and biomass burning carbon emissions over specific regions (63-90°N, 120-170°E) for the period of 2002-2021. (a) ALT from the CCI-PF (unit: m), (b) soil moisture over 0-7 cm depth from the ERA5 reanalysis (unit: m³/m³), and (c) biomass burning carbon emissions from the GFEDv4.1 (unit: TgC).

Another concern relates to the projected abrupt increase in evapotranspiration (fig. S7). The authors attribute the abrupt increase in evapotranspiration “due to an increase in photosynthetic carbon uptake, which is determined by both CO₂ fertilization and warming effects”. However, both warming and CO₂ fertilisation are unlikely to change abruptly in the 21st century and, thus, the change in evapotranspiration is equally unlikely. Moreover, plant evapotranspiration should decrease as a result of low soil moisture. It is important to understand the reason behind the abrupt increase in evapotranspiration because this variable is linked with fuel moisture and, hence, fire risk.

We agree with the reviewer’s assertion that the increase in photosynthesis caused by CO₂ fertilization is unlikely to be abrupt in the 21st century. However, the effect of CO₂ fertilization contributes to an increase in photosynthesis, which can subsequently lead to an increase in evapotranspiration.

The CESM2-LE shows that the abrupt increase in evapotranspiration is consistent with the timing of an abrupt decrease in relative humidity (see Fig. R2). This result suggests that the rapid increase in canopy evapotranspiration over ice-rich permafrost regions is mainly caused by an increase in atmospheric water demand resulting from rapid atmospheric drying^{14,15} (Seneviratne, S. I. et al., 2010, Helbig, M. et al., 2020).

Our revised text now includes a discussion of the comments brought up by the reviewer (see blue revised text in main manuscript).

Lines 170-175: Furthermore, drier atmospheric conditions can increase atmospheric water demand, leading to an increase in canopy evapotranspiration. Additionally, the effect of future CO₂ fertilization can enhance vegetation growth, which further amplifies evapotranspiration. The rapid increase in canopy evapotranspiration mainly occurs over areas where soil ice melts quickly, which is consistent with the timing of abrupt soil drying and atmospheric drying (Fig. 3, Figs. S5-7).

Figure R2 Time evolution of relative humidity and evapotranspiration over the permafrost regions among 50 ensemble members. (a) 65.5°N, 72.5°E; (b) 65.5°N, 83.75°E; (c) 55.13°N, 85.0°W; (d) 57.02°N, 92.5°W. Brown indicates relative humidity (units: %) and green indicates canopy evapotranspiration (units: mm/month). Solid lines represent one ensemble member, and shading indicates ± 1 standard deviation of 50 ensemble members.

The Discussion section falls short of engaging with the existing literature. The Discussion section only includes three references, none of which were used to contrast the findings of the study. By citing previous studies, the authors have an excellent opportunity to further support the model's projections. For instance, 1) it is known that soil moisture, more concretely lowering the water table, can increase the extent of peat fires (Turetsky, M. et al., 2015). 2) Other studies, included in the Introduction section, explain the changes in hydrology, supporting the model's outputs. 3) The model projections could be compared with future projections in burned areas obtained in previous studies.

Following the reviewer's suggestion, the discussion section has been complemented in the revised manuscript (see blue revised text in main manuscript).

Two paragraphs were placed in the incorrect section of the manuscript. First, the last paragraph of the Introduction (L62-L70) presents results about model validation. This sentence belongs to the Results section rather than the Introduction. Also, please note that, according to the formatting guideline of the journal, "The final paragraph must begin with a phrase like "In this work" or "Here, we show", and contain a summary of the major results and conclusions of the current work, written in the present tense.". The Discussion section also presents additional results in L221-224 and L228-236. This paragraph also belongs to the Results section.

Following the reviewer's suggestion, the last paragraph of the Introduction has now been corrected in the revised version (see blue revised text in main manuscript).

Lines 222-225: In this work, our analyses of 50 ensemble simulations of the CESM2-LE under historical/SSP3-7.0 forcing demonstrated that permafrost thawing in the Subarctic/Arctic region can serve as a trigger for abrupt regime shifts in soil hydrological processes and regional wildfires (Fig. 6).

However, we have kept the L221-224 and L228-236 in the discussion section for the following reasons.

In L221-224, we aim to discuss the potential contribution of lightning ignition to future wildfire activity with the earlier study, as our study does not consider the response of lightning to climate change for wildfire activity (See Lines 265-275). In L228-236, the discussion of the contribution of wildfires to the terrestrial carbon cycle provides the practical implications of the main finding (see Lines 279-287).

Lines 231-259: The simulated abrupt increase in wildfires following rapid permafrost thawing is consistent with the findings of an earlier study, which analyzes the atmospheric Fire Weather Index. In contrast to this study, our CESM2-LE simulates the interactions between climate-vegetation-permafrost and fires explicitly which leads to a different representation of important coupled feedbacks and dynamics. In the CESM2-LE simulation, subgrid-scale permafrost processes are parameterized in such a way that permafrost thawing in certain areas leads to a subsequent abrupt soil drying. This may be an oversimplification of the scale-dependent dynamics that can occur in geographically diverse permafrost regions. Polygonal permafrost landforms at the meter-scale can significantly influence the hydrological cycle even at the watershed-scale, but the representation of multi-scale interactions is one of the key challenges in ESMs. There have been new modelling developments to improve the representation of permafrost dynamics. For instance, experiments using the new parameterization with the Community Land Model 5 (CLM5) suggest that subsidence due to permafrost thaw can increase the surface water fraction. In addition, small-scale simulations of the ice-rich lowlands have shown that thaw subsidence under waterlogged conditions can increase soil water saturation, thereby accelerating thaw, whereas under well-drained conditions soil water saturation decreases. It is important to note that incorporating these new modelling developments may lead to results that differ from the hydrological responses to permafrost thaw in our study. Therefore, for future studies using ESMs, it is essential that consider these recent modelling advancements and compare them with existing models.

To date there is little direct observational evidence supporting that permafrost thaw-induced soil drying leads to an increase in wildfires on a large scale, as suggested by the CESM2-LE. However, observations show that permafrost thawing can lead to drying of surface water bodies and that atmospheric warming and increasing aridity, and decreasing ground moisture contribute to wildfire activity. Other supporting evidence for some of the key processes highlighted in our study comes from studies on northern peat bogs, which show that drainage can increase carbon emissions due to peat burning and that lower water tables under a warmer climate can potentially enhance the risks of peat burning.

Specific comments:

L15: The title and abstract only mention the subarctic region. However, results and figures also include the Arctic region.

This has now been corrected in the revised version (see blue revised text in title and abstract).

L17-18: The causal relation of permafrost thawing on increased wildfire activity has not been demonstrated. The authors have pointed this out in L43, as part of the justification of the study. Thus, the statement L17-18 requires rephrasing.

This has now been corrected in the revised version (see blue revised text in main manuscript).

Lines 18-19: The impact of permafrost thawing on future Arctic-Subarctic wildfires and the associated release of greenhouse gases and aerosols is less well understood.

L26: Instead of mentioning 'several orders', specify the order or range of magnitude of the abrupt intensification.

This has now been corrected in the revised version (see blue revised text in main manuscript).

Lines 25-27: These processes combined, lead to nonlinear late-21st-century regime shifts in the coupled soil-hydrology system and a rapid intensification of wildfires in northern Siberia and Canada.

L29: 'wet environment' might mislead the reader. Although evapotranspiration is low, the tundra biome

receives low levels of precipitation and, thus, could be also labelled as a dry environment.

This has now been corrected in the revised version (see blue revised text in main manuscript).
Lines 29: The cold environment of the Arctic is typically associated with long climatological fire return intervals.

L35-39: 'low soil moisture' suits better than 'arid soils' in this sentence. Also, I suggest replacing 'soil wetness' by 'soil moisture' to keep a consistent terminology throughout the manuscript.

This has now been corrected in the revised version (see Lines 33-36 in main manuscript).
Lines 33-36: Fire occurrences in the Arctic and Subarctic are in part controlled by the immediate atmospheric conditions (fire weather), and in part by soil water conditions, with arid soils increasing the frequency, and spatial extent of fires, and by the availability of fire fuel.

L35: Please check whether Ref. #6 actually supports the statement 'arid soils increase the probability, frequency, and spatial extent of fires.'

This has now been corrected in the revised version (see Lines 33-36 in main manuscript).
Lines 33-36: Fire occurrences in the Arctic and Subarctic are in part controlled by the immediate atmospheric conditions (fire weather), and in part by soil water conditions, with arid soils increasing the frequency, and spatial extent of fires, and by the availability of fire fuel.

L53 mentions 'previous studies' but only one reference is provided.

This has now been corrected in the revised version (see Line 51 in main manuscript).

L78-80: The maps in Fig. S3a,b does not show the association between warming and ALT in a clear manner. In fact, although the southern margin of the permafrost region presents a pronounced trend in ALT, this region presents a warming trend that is lower than the northern part of the permafrost region. I suggest a scatter plot between warming trends and ALT trends to elucidate the association.

The explanation in L78-80 has been corrected in the revised version (see blue revised text in main manuscript).

Lines 69-72: The increasing trends in ALT are particularly pronounced along the southern edge of the permafrost zone (Fig. S3c), where the thawing threshold will be crossed more easily due to higher climatological summer temperatures as well as in northwestern Siberia [60-80°E, 65-70°N].

L89: Does 'anthropogenic effects' refer to 'anthropogenic warming'?

This has now been corrected in the revised version (see blue revised text in main manuscript).
Lines 79-80: Comparing natural variability with projected trends from 1997-2019 in temperature and ALT reveals that the anthropogenic greenhouse warming (as represented by the ensemble mean or mode) in the region under consideration are already emergent above the internal variability (Figs. S3d-f), at least beyond the interquartile range (IQR).

L115-116: Could rapid permafrost thawing increase soil moisture in flat terrains and depressions? If that is the case, the increase in fire risk could be highly spatially dependent. I wonder if the model can make predictions based on geomorphology.

There have been recent efforts to develop permafrost thaw-associated land surface subsidence and hydrological changes to more realistically represent thermokarst processes in Earth System Modeling. Ground subsidence induced by the melting of excess ice can increase surface water fractions¹⁶ (Ekici, A. et al., 2019). In a warmer climate, permafrost thawing in poorly drained areas over the Siberian lowlands can lead to the formation of surface water bodies at a land-scape scale¹⁷ (Nitzbon, J. et al., 2020). The increased soil moisture caused by thawing may potentially reduce fire risk. However, the

hydrological coupling between permafrost thaw-induced subsidence and surface inundation has not been incorporated in the CESM2-LE. Therefore, this study does not account for the effects of ground subsidence on permafrost-related hydrological processes at a sub-grid scale.

L118-122: The justification for focusing on western Siberia is not sufficient. The same justification could be used for selecting another region in the arctic or subarctic. At the very least, these analyses should also focus on a fire-prone region (for instance, Sakha republic) and compare the trends in both regions. Ideally, the study should subdivide the arctic and subarctic in subregions and analyze the trends in each subregion to provide a comprehensive overview of future projections.

We have added the analysis for fire-prone region in the revised version (see Figs. S6-7 and blue revised text in main manuscript).

Lines 136-140: The temporal evolutions in other permafrost locations show that a rapid decrease in soil ice mainly occurs in ice-rich areas, which is consistent with sudden shifts in sub-surface runoff and soil moisture from the mid-to-end of the 21st century (Fig. S6). In contrast, in regions with less soil ice, the reduction in soil ice occurs more gradually after the early of 21st century, thereby sustaining sub-surface runoff and upper soil moisture levels (Fig. S7).

Lines 191-195: Furthermore, the sudden increase in wildfires occurs primarily after sudden thaw-induced soil drying over ice-rich permafrost regions (Fig. S6). In contrast, there is no abrupt increase in wildfires in a warmer climate over historically fire-prone regions near the southern edge of the permafrost domain, which can be explained by the absence of abrupt changes in soil ice and soil moisture (Fig. S7).

L122: The authors must clarify the month that was selected for these results. According to the caption in Fig. 1, these results were obtained for October. Also, the manuscript must explain why October was selected for this analysis while July was selected in other figures.

In observation, the ALT is defined as the maximum thaw depth in order to monitor permafrost. Therefore, we have made a modification, from the October ALT to the maximum annual ALT (see Figs.1a,b,h).

L134-139: These results might be substantially different in other regions of the arctic. Consider including more subregions for this analysis.

Following the suggestion of the reviewer, we have added the analysis for other locations within the permafrost domain in the revised version (see Figs. S6-7 and blue revised text in main manuscript).

Lines 173-175: The rapid increase in canopy evapotranspiration mainly occurs over areas where soil ice melts quickly, which is consistent with the timing of abrupt soil drying and atmospheric drying (Fig.3, Figs.S5-7).

L148-153: Following previous comment, Fig. 3 and Fig. 4 show that the change in fire regime will only occur in certain regions on the arctic and subarctic. The manuscript should clarify that the projected abrupt soil drying and subsequent increase in fire activity will occur only in these specific regions.

We now clarify that the projected abrupt soil drying and subsequent increase in fire activity will occur over the specific regions in the revised version (see Figs. S6-7, Lines 136-140, and Lines 191-195).

Moreover, according to Fig. 3a, projected evapotranspiration will increase in historical fire prone regions of the subarctic. Given the direct link between evapotranspiration and fuel moisture, these results might be interpreted as vegetation will be less susceptible to fire.

Locally increased evapotranspiration could lead to increased precipitation, thereby making the region less prone to fire. However, the positive relationship between evapotranspiration and precipitation is an uncertain link in the soil moisture-precipitation feedback¹⁴ (Seneviratne, S. I. et al., 2010).

In the CESM2-LE, evapotranspiration (EVT) increases significantly over the historically fire-prone region, while the change in precipitation and is relatively small (see Figs. R3a, b). In addition, changes in near-surface soil moisture and burned area are stabilized over the historically fire-prone region (62.67°N, 122.5°E) in a warmer climate. Although, there is an increase in EVT in a warmer climate, it's difficult to establish a clear causal relationship between EVT and burned area in Fig. R3. From Fig. R3, we can interpret that the key factor in reducing fire susceptibility in this region is sustained near-surface soil moisture.

Figure R3 Time evolution of gradual changes over the regions (62.67°N, 122.5°E) among 50 ensemble members. (a) total evapotranspiration (unit: mm/month), (b) sum of rainfall and snowfall (unit: mm/month), (c) soil moisture over 0-10cm depth (unit: kg/m²), and (d) burned area (unit: km²), bold lines indicate the ensemble mean and thin lines indicate 50 ensemble members.

L161-162: The terms ‘abrupt’ and ‘rapid’ increase/decrease are used subjectively. Surface temperature also shows a significant change comparable to ground heat.

This has now been corrected in the revised version (see blue revised text in main manuscript).

Lines 165-166: Along with these changes, the ensemble mean of surface air temperature increases considerably (2040:15.4±3.0°C, 2060:18.5±2.8°C) (Fig. S8b).

L172: I suggest providing changes in burned area in absolute terms (km²) or density of burned area (km² burned per pixel size) instead of percentages. The >1000% might mislead the reader. This extreme change might be only possible because western Siberia and eastern Canada present no current fire activity.

Following the reviewer’s comments, we have changed the units from relative changes (percentage change) to absolute changes with a logarithmic scale (see Fig. 4).

L177-179: Following previous comment, the changes in the burned area in absolute terms seem unrealistic. According to Fig. 4c, the burned area will increase up to 1000 km² only in a pixel in Western Siberia by the end of the century. Information on the pixel size is key to understanding this figure panel.

In the CESM2-LE, the size of the grid cell over this region (65.5°N, 83.75°E) is 6,041km². On average, the annual burned area will reach approximately 800km² in this region by the end of the 21st century. In other words, around 13% of the grid cell in this region this region will be affected by burning by the end of the 21st century.

Reviewer #2 (Remarks to the Author):

Summary

The authors used a large ensemble simulation performed by the state-of-the-art Earth system model which considers the coupling between permafrost and hydrology. They investigated the relationship between 1) permafrost thawing, 2) thawed water percolation to a deeper layer, 3) soil drying at the upper layer, 4) an increase in surface air temperature and decrease in relative humidity, and 5) an increase in wildfire occurrences. I think this topic is very important and interesting, but I believe the conclusions are not supported by strong evidence. Therefore, I should say that the current manuscript does not satisfy the criteria for the publication of Nature journals. The reasons are explained as follows. I recommend that the authors consider these points and improve the manuscript.

We thank the reviewer #2 for the constructive comments. We have revised the manuscript in response to the reviewer's concerns.

General Comments

1. First, the causal relationships from the above 1) to 5) are shown mainly based on a “representative” grid point in northwestern Siberia (65.5N, 83.75E). Therefore, I couldn't really understand what was happening elsewhere. The authors show the maps of changes in active layer thickness, soil ice, soil moisture, runoff, evapotranspiration, Bowen ration, and burned area by performing the “change point analysis” (Figure 1-4). I am not sure that the changes in these variables are really caused by permafrost thawing, because the map of ALT changes (Figure 1a) is not consistent with other variables (Figure 1c, 1e, 2b, 3b, 3d), and similarities and differences between these distributions are not discussed in the manuscript. In the first place, will the permafrost thaw rapidly (abruptly) at any location? If the changes in permafrost and other variables are gradual and not abrupt, what do the results of “change point analysis” mean? This analysis assumes that “one abrupt change occurs over each region” (Line 295). In fact, the distribution in Figure 1a (the timing of ALT change) appears implausible. The timing of changes in the Arctic Ocean coastline at higher latitudes is earlier than in inland areas at lower latitudes. Furthermore, the distribution of ALT appears to be uniform.

As the reviewer points out, we presented the analysis for a representative grid point in northwestern Siberia (65.5°N, 83.75°E). Following the reviewer's comment, we have included analyses for other permafrost locations (see Figs 6-7 and Lines 136-140 in main manuscript). We have also included an explanation of the similarity between ALT and deeper soil ice distributions (see Lines 89-92).

Lines 89-92: The timing of ALT and deeper pore ice tends to be similar over western Siberia and Canada. The rapid changes in ALT and deeper pore ice overall tend to occur earlier over the lower latitudes except for the regions near Arctic coastlines.

Lines 136-140: The temporal evolutions in other permafrost locations show that a rapid decrease in soil ice mainly occurs in ice-rich areas, which is consistent with sudden shifts in sub-surface runoff and soil moisture from the mid-to-end of the 21st century (Fig. S6). In contrast, in regions with less soil ice, the reduction in soil ice occurs more gradually after the early of 21st century, thereby sustaining sub-surface runoff and upper soil moisture levels (Fig. S7).

The reviewer questions whether the changes in the variables (e.g., soil moisture and runoff etc.) are really caused by permafrost thaw. Our results clearly show that significant changes in deeper soil ice are linked to the changes in upper soil moisture and burned area (Fig.1, Figs.4d-e, and Fig. S6). However, there may be inconsistencies between the changes in ALT and other variables (e.g., deeper soil ice etc.) in some regions. This discrepancy is attributed to ALT being determined by the soil temperature over the entire soil layer (0-40m) in the model, rather than focusing on hydrological properties in the deeper layer.

The reviewer points out our consideration (“one abrupt change occurs over each region”). In our study, we consider the case where only one or no abrupt transition could occur in each region in order to focus on major changes over 1850-2100 (see Lines 357-358). The change point analysis algorithm detects one structural change at each grid point during the period 1850-2100. It is therefore limited in its ability

to clearly exclude some gradual changes. However, our focus is primarily on larger changes in soil ice, especially those greater than 50%, during the 20-year period between pre- and post-thaw (Fig. 1d). We believe that this represents the rapid changes in permafrost thaw (Figs. 1-4, Fig. S6).

The reviewer points out that the changes in the Arctic Ocean coastline at higher latitudes occur earlier than in inland areas at lower latitudes. Although the changes in ALT occur earlier in regions close to Arctic coastlines than in inland areas at lower latitudes, the magnitude of the changes is relatively much smaller compared to other permafrost regions.

In addition, the reviewer comments on the uniform spatial distribution of ALT compared to soil ice and soil moisture. This arises because ALT is determined by changes in the spatially uniform soil temperature in the model.

The authors should explain the changes in permafrost thawing and other variables should be abrupt at any location (and what determines the time scale of thawing) because they assume “one abrupt change occurs over each region” in their analysis. In addition, the readers would be interested in the time sequence at locations other than the representative grid cell. The authors should show the time sequence with regional average for different regions.

We have revised an explanation of the change point analysis (see Lines 357-358) and the analysis for the other locations in the permafrost domain. For the analysis in the other locations, we consider relatively ice-rich and ice-poor permafrost regions (see Lines 136-140 and Figs. S6-7).

Lines 357-358: To examine the largest change in each region during 1850-2100, we consider the case where only one or no abrupt change could occur in each region.

Lines 136-140: The temporal evolutions in other permafrost locations show that a rapid decrease in soil ice mainly occurs in ice-rich areas, which is consistent with sudden shifts in sub-surface runoff and soil moisture from the mid-to-end of the 21st century (Fig. S6). In contrast, in regions with less soil ice, the reduction in soil ice occurs more gradually after the early of 21st century, thereby sustaining sub-surface runoff and upper soil moisture levels (Fig. S7).

2. I could understand the causal relationship from 1) to 4) at the representative grid cell (65.5N, 83.75E). However, the relationship between the permafrost thawing and 5) increase in wildfire occurrence is not shown with strong evidence. The author claims that wildfires increase from 2050 (Figure 4c), but it is unclear whether this is due to the permafrost thawing or changes in atmospheric conditions. In addition, the wildfire occurrence increases as shown in Figure 7b, but the role of permafrost thawing in Figure 7b is not clear (although it is the result of permafrost domain, changes in atmospheric conditions could increase the wildfire over there). The wildfire increases in the sensitivity experiments (Figure 5), but the experimental setting, i.e., abrupt 40% (80%) reductions within all layers over the high latitudes (> 40 N), should not correspond to the conditions after permafrost thawing (the authors should explain why they chose these experimental settings).

The reviewer points out that it is unclear whether the increase in wildfires from 2050 is due to permafrost thaw or changes in atmospheric conditions. In the CESM2-LE, the abrupt increase in Arctic and Subarctic wildfires can be attributed to permafrost thaw or changes in atmospheric conditions.

Therefore, to support our suggestion that the abrupt soil drying following permafrost thaw is the main driver of the abrupt increase in wildfires from the CESM2-LE, we additionally conduct the idealized experiments with perturbations of soil moisture reduction and compare them with the control simulation (see Lines 342-351 in the revised manuscript).

The control simulation represents the soil moisture state prior to permafrost thaw. In contrast, the idealized experiments are designed to include 20% and 40% soil moisture reduction perturbations at higher latitudes (>40°N) to reflect the extent of soil moisture reduction after permafrost thaw in the CESM2-LE. The comparison between the control simulation and the soil moisture reduction experiments allows to identify the effects of abrupt soil drying on wildfires in permafrost regions. The idealized experiments clearly show that the abrupt decrease in soil moisture induces the significant

atmospheric conditions, including an increase in surface air temperature and a decrease in relative humidity, that lead to an increase in wildfires (see Fig. 5).

In addition, the reviewer notes that the cumulative carbon release from wildfires in Fig. 7b can be influenced by atmospheric conditions as well as permafrost thaw. However, Fig. 7b mainly includes changes in the cumulative carbon release from wildfires following permafrost thaw. Therefore, we believe that quantifying the cumulative carbon loss from wildfires in permafrost regions, where abrupt wildfires occur, provides a rough estimate of the contribution of wildfires to the terrestrial carbon balance of these regions in a warmer climate.

3. The authors discuss the future projection of permafrost and hydrology based on their simulation results, but they should consider other possible processes not described in CESM2. In the literature of future permafrost projections, the increase in methane emissions by wetland expansions is discussed. Are there any areas where the wetlands are expanding in CESM2? Is the “abrupt drying” after permafrost thawing in CESM caused by the model not considering the more complicated phenomena (e.g., thermokarst formation)? I understand it is not easy to consider the complicated phenomena in the global model, but I hope the possibility it should be discussed in the paper.

Following the reviewer’s suggestion, we have complemented the discussion of the expansion of wetlands (see Lines 236-251). The CESM2-LE output includes a representation of the areas with ground covered by surface water (FH2OSFC), but the CESM2-LE does not incorporate thermokarst dynamics. In the CESM2-LE, FH2OSFC decreases significantly (>50%) over western Siberia and Canada during 2080-2100 compared to 1995-2014 (see Figs. R4a,b), consistent with upper soil drying (Fig. 1f). However, within a specific region (65-70°N, 110-130°E), wetlands (FH2OSFC) are likely to stabilize under a warmer climate (Fig. R4c). It should be noted that this parameter (FH2OSFC) is not used to estimate CH₄ production and oxidation in the CESM2-LE. Therefore, it is not easy to represent the natural phenomena of thermokarst formation with FH2OSFC. Thermokarst dynamics are important to better understand permafrost-carbon-climate feedbacks, and model development efforts are currently underway in the CLM model¹⁶ (Ekici, A. et al., 2019) (see blue revised text in the main manuscript).

Lines 236-251: In the CESM2-LE simulation, subgrid-scale permafrost processes are parameterized in such a way that permafrost thawing in certain areas leads to a subsequent abrupt soil drying. This may be an oversimplification of the scale-dependent dynamics that can occur in geographically diverse permafrost regions. Polygonal permafrost landforms at the meter-scale can significantly influence the hydrological cycle even at the watershed-scale, but the representation of multi-scale interactions is one of the key challenges in ESMs. There have been new modelling developments to improve the representation of permafrost dynamics. For instance, experiments using the new parameterization with the Community Land Model 5 (CLM5) suggest that subsidence due to permafrost thaw can increase the surface water fraction. In addition, small-scale simulations of the ice-rich lowlands have shown that thaw subsidence under waterlogged conditions can increase soil water saturation, thereby accelerating thaw, whereas under well-drained conditions soil water saturation decreases. It is important to note that incorporating these new modelling developments may lead to results that differ from the hydrological responses to permafrost thaw in our study. Therefore, for future studies using ESMs, it is essential that consider these recent modelling advancements and compare them with existing models.

Figure R4 Ground covered by surface water in CESM2-LE over the Arctic and Subarctic regions. (a) Annual ground covered by surface water (FH2OSFC) during the period 1995-2014 (unit: km²), (b) differences in ground covered by surface water between the period of 1995-2014 and 2081-2100, (unit: %), and (c) temporal evolution of annual ground covered by surface water over the regions (65-70°N, 110-130°E) during the period 1850-2100 (unit: km²).

Individual comments

Line 24: As I commented above, the authors show the results based on a single grid point. I am not sure if these results show the regime shifts over the subarctic.

In the revised version, we have supplemented the original analysis from the submitted manuscript version with analysis for other locations within the permafrost domain (see Figs 6-7, Lines 136-140, and Lines 191-195 in main manuscript).

Line 62: This paragraph is the results of model simulations, and thus it is not suitable for the introduction.

This paragraph has now been moved to the Methods section in the revised version (see Lines 331-339).

Line 100: “Historical permafrost regions (> 70°N)”. Do you refer to “permafrost regions” as the entire region over 70°N? The phrase “permafrost regions” is used in the manuscript, but they are not consistent (for example, > 55°N in Figure 7). Why you change the region? The term “permafrost” is usually defined as the region where the ground temperature is below 0 degC for at least two years. The authors should define this term and explain why they chose different regions. The definition of “historical permafrost domain” is (probably, only) shown in the caption of Supplementary Figure 4 (Line 40 in SI) as “the area where ALT is less than 3m”, but why the authors choose this threshold (ALT < 3 m)?

“Historical permafrost regions (< 70°N)” refers to the areas located south of 70°N within the historical permafrost region. Our explanation in line 100 may have caused confusion, so we have corrected the text in Lines 100 and Fig. 7 and have added an explanation defining of the permafrost area in the caption of Fig. 1.

In our study, the historical permafrost region is defined as the area where ALT is less than 3m during the period of 1850-1869. Previous studies using Earth System Models (ESMs) have defined the permafrost region where ALT is less than 3m in order to focus on the near-surface permafrost¹⁸⁻²⁰ (Dankers, R. et al., 2011, Chen, Y. et al., 2023, Peng, X. et al., 2023, because it represents the depth range where permafrost significantly influences hydrological and biogeochemical processes.

Lines 92-94: In terms of the magnitude of the forced changes (Figs. 1b,d,f) we find a substantial increase in ALT over most of the historical permafrost regions exceeding 3m (Fig. 1b).

Line 105: “abrupt increase in ALT”. As shown in Supplementary Figure 3f, the trend in ALT is about 0.1 m/23yr. On the other hand, the increase in ALT shown in Figure 1h is about 40 m during about 20 years. Are these results in Supplementary Figure 3f and Figure 1h consistent? Is the ALT increase shown in Figure 1h physically reasonable (what determines the time scale of permafrost thawing)?

In CLM5, the ALT is defined as the depth at which the soil temperature reaches 0°C throughout the entire soil layer in the model (0-40m). The depiction of 40m in the previous Figure 1 did not account for bedrock. Despite the model encompassing a soil depth of 40m, our study focuses on changes in near-surface permafrost. The Fig. 1h has now been updated to include bedrock considerations in the revised version (see Fig. 1h).

Line 107: “The timing of the 0 degC 10 cm ground temperature exceedance serves as a good proxy for the abrupt responses in soil ice and soil water”, but why is the timing of “10 cm ground temperature exceeds 0 degC” so important?

The temporal evolution over Western Siberia (65.5°N, 83.75°E) shows that, after the annual mean soil temperature in 0-10cm depth reaches 0°C, delayed thawing occurs in deeper soil layer by gradual downward heat transfer (see Figs. 1-2).

Line 128: If you can show the results of “vertical hydraulic conductivity”, it is helpful to support the hypothesis. I am interested in how they change during the simulation.

Vertical hydraulic conductivity in the deeper layer (>3m) increases over western Siberia (65.5°N, 83.75°E) after 2050, consistent with a decrease in upper soil moisture (see Fig. R5 and Swenson, S. C. et al., 2012²¹).

Figure R5 Vertical hydraulic conductivity and volumetric soil water over Western Siberia (65.5°N, 83.75°E) during the period 2000-2100. Depth-time cross-sections of (a) vertical hydraulic conductivity (unit: m) and (b) volumetric soil water (unit: m³/m³).

Line 134: “Transpiration and evaporation from the plants increase ... due to increase in photosynthetic carbon uptake, ...”. This is interesting. I think the CO₂ fertilization effect reduces transpiration by the changes in stomata. Can you add more explanation?

As the reviewer mentions, drought stress can cause a decrease in transpiration due to lower stomatal conductance under higher CO₂ concentrations. Conversely, transpiration can also increase due to atmospheric water demand^{14,15} (Seneviratne, S. I. et. al., 2010, Helbig, M. et al., 2020).

In the CESM2-LE, an increase in vegetation resulting from the CO₂ fertilization effect can increase evapotranspiration (EVT) in Western Siberia (65.5°N, 83.75°E). Furthermore, the rapid increase in EVT is accompanied by a rapid decrease in relative humidity (Fig. R2), suggesting that drier atmospheric conditions contribute to an increase in EVT.

We have complemented the explanation on line 134 in the revised version (see Lines 170-175 in the main manuscript).

Lines 170-175: Furthermore, drier atmospheric conditions can increase atmospheric water demand, leading to an increase in canopy evapotranspiration. Additionally, the effect of future CO₂ fertilization can enhance vegetation growth, which further amplifies evapotranspiration. The rapid increase in canopy evapotranspiration mainly occurs over areas where soil ice melts quickly, which is consistent with the timing of abrupt soil drying and atmospheric drying (Fig. 3, Figs. S5-7).

Line 163: “This can intensify surface warming, thus leading to a rapid decline in relative humidity ... despite a smaller change in the actual amount of water vapour”. This explanation is difficult to follow. Is the decline in relative humidity simply caused by 1) surface warming due to less latent heat and 2) less water vapour due to less evaporation?

The abrupt decrease in relative humidity is influenced by an increase in surface air temperature resulting from an abrupt decrease (increase) in latent heat flux (sensible heat flux). It is also reflected in a decrease in actual water vapor, but the contribution of actual water vapor is relatively small.

Line 185: As I described above, why did the authors choose the abrupt decrease by 40 or 80 % in regions poleward of 40°N? The reason why you chose these settings should be explained. Otherwise, we cannot understand the meaning (how it is related to the realistic future projection) of the idealized experiments. To show the impact of permafrost thawing, the experiments starting from the “post-thaw” conditions can be useful to show the causal relationship between permafrost thawing and the increase in wildfire occurrence, for example.

We have commented in our responses above to the comments about the idealized experiments. In the revised version, we have provided additional details to improve the explanation of our experimental settings. (see Lines 342-347).

Lines 342-347: In the two idealized experiments, we impose artificial reductions in soil moisture content within all soil layers by 20% and 40% over the high latitudes (>40°N) from the July 1st, 2045 model state (pre-thawing condition) in one ensemble member of the CESM2-LE. The choice of 20% and 40% reduction in soil moisture for the idealized experiments is made to account for the abrupt 20-40% reduction in soil moisture following permafrost thawing in the CESM2-LE.

Line 234: “Cumulative net uptake by ecosystem production (NEP) over the same permafrost regions (2.03±0.51 PgC)”. It seems small compared to the annual land carbon uptake is reported about 3 GtC per year (<https://essd.copernicus.org/articles/14/4811/2022/>). Is this number, correct?

The reason why our results are smaller is that we only consider the NEE over the permafrost regions where abrupt changes in wildfire occur. We have revised the line 234 and Fig. 7 with the revised the definition of permafrost region (see Lines 279-283 and Fig. 7).

Lines 279-283: Our estimation from the CESM2-LE shows that wildfires occurring in permafrost regions where the abrupt changes occurred would cumulatively release 1.40±0.26 PgC towards the end

of the 21st century, the absolute value of which amounts to about 64% of cumulative net uptake by ecosystem production over the same permafrost regions (-2.18 ± 0.53 PgC) (Fig.7b).

Line 243: As commented above, the thermokarst formation and wetland expansions would be also an important factor in projecting the future permafrost and hydrological change.

We have complemented the discussion in the revised version (see Lines 236-251).

Figure 1: In some regions along the arctic coast, abrupt change in ALT is earlier than the lower latitudes. I expect that the time of abrupt change in the higher latitudes become later than the lower latitudes.

Although the rapid changes in ALT occur earlier in regions near Arctic coastlines than in inland areas at lower latitudes, the magnitude of the changes is relatively much smaller than in other permafrost regions.

Figure 3: Only this figure shows the difference between pre- and post-thaw. This seems reasonable to show the impacts of permafrost thawing. Other maps (Figures 1, 2, and 4) are calculated as the pre-and post-abruptness: as I commented above, the results are difficult to understand.

Figs.1, 2, and 4 illustrate the occurrence of abrupt changes over permafrost regions. To complement the explanation of the effects of permafrost thaw on wildfires, we have updated Fig. 4 and Figs. S6-7, S9 in the revised version.

Figure 7b: Since this is cumulative carbon emission, the values of “Fire” and “GFED4.1s” should be positive (negative means decrease?). I also noticed a large difference between the model simulations and GFED. This should be discussed in the manuscript (even though the difference between the simulations and observation averaged over 1997-2004 are discussed in Line 66).

Fig. 7b has now been corrected in the revised version. We have also complemented an additional explanation for the large difference between the model simulation and GFED in the revised manuscript (see Fig. 7 and Lines 260-264).

Lines 260-264: Some other modelling caveats need to be mentioned: The CESM2-LE underestimates the observed burned area over the Arctic and Subarctic regions compared to tropical and temperate latitudes (Fig. S2). This could be due to the model’s lack of explicit representation of changes in fire ignition. The model uses a fixed climatological lightning frequency for natural ignition without internal lightning noise.

Reviewer #3 (Remarks to the Author):

The authors present a comprehensive analysis of some of the effects of future permafrost thawing on land surface processes, with a particular focus on the interactions between permafrost, soil hydrology and wildfire. Their analysis is based on an ensemble of 50 simulations performed using a state-of-the-art Earth system model, which includes permafrost-related soil thermal and hydrological dynamics, carbon cycle dynamics, and a process-based fire parameterization. The manuscript is well organized, easy to follow, and effectively uses high-quality graphics to illustrate the main points of the analysis.

The manuscript is consistent with previous research which showed that permafrost thawing can lead to surface soil drying, and that this drying affects upward and downward fluxes of heat and moisture. As acknowledged in the manuscript, the conclusion that permafrost thawing-led soil drying has the potential to increase wildfire activity was first presented in Teufel and Sushama, 2019 (hereafter, TS19). It is important to note that the model used for the current study incorporates carbon cycle dynamics, including a process-based fire parameterization, which was not the case for TS19. Including the carbon cycle enables the analysis to account for future changes in vegetation, and the fire parameterization allows for the explicit quantification of fire counts, burned area and fire type (e.g., biomass vs peat). In addition, the ensemble of 50 members in the current study (10 times larger than in TS19) increases the statistical robustness of the conclusions.

Permafrost landscapes are complex, and some of this complexity cannot be captured at Earth system model scale (~100 km horizontal resolution). For example, subgrid orography will have a strong influence on moisture pathways as permafrost thaws, which implies that some parts of a grid cell might experience wetter soil conditions, while others dry out. This complexity does not invalidate the conclusions of the study, as many regions are projected to experience drying, and the ensuing consequences detailed in the manuscript. However, it would be valuable for this complexity to be discussed in the manuscript and to acknowledge that the response at local scales might be significantly different from the CESM2 projections.

We thank the encouraging comments of the manuscript provided by the Reviewer #3. We believe that the reviewer's concerns about the discussion of the complexity of permafrost landscapes have been appropriately addressed in the updated version of the manuscript (see blue revised text in Lines 236-251 in main manuscript).

Lines 236-251: In the CESM2-LE simulation, subgrid-scale permafrost processes are parameterized in such a way that permafrost thawing in certain areas leads to a subsequent abrupt soil drying. This may be an oversimplification of the scale-dependent dynamics that can occur in geographically diverse permafrost regions. Polygonal permafrost landforms at the meter-scale can significantly influence the hydrological cycle even at the watershed-scale, but the representation of multi-scale interactions is one of the key challenges in ESMs. There have been new modelling developments to improve the representation of permafrost dynamics. For instance, experiments using the new parameterization with the Community Land Model 5 (CLM5) suggest that subsidence due to permafrost thaw can increase the surface water fraction. In addition, small-scale simulations of the ice-rich lowlands have shown that thaw subsidence under waterlogged conditions can increase soil water saturation, thereby accelerating thaw, whereas under well-drained conditions soil water saturation decreases. It is important to note that incorporating these new modelling developments may lead to results that differ from the hydrological responses to permafrost thaw in our study. Therefore, for future studies using ESMs, it is essential that consider these recent modelling advancements and compare them with existing models.

Other comments:

Title can lead to confusion. The current title: "Abrupt increase in subarctic wildfires amplified by future permafrost thawing" can be interpreted to imply that an abrupt increase in wildfires would occur in all cases, and that permafrost thawing only serves to amplify said abrupt increase. The abstract and manuscript make it clear that the abrupt increase in wildfires is actually a consequence of permafrost

thawing, clarity that should also be reflected in the title.

We have revised the title (see Line 1).

Line 1: Abrupt increase in Arctic-Subarctic wildfires caused by future permafrost thawing”.

L18: "which can also contribute" should be "which also contribute".

This now been corrected in the revised version (see blue revised text in main manuscript).

Lines 18-19: The impact of permafrost thawing on future Arctic-Subarctic wildfires and the associated release of greenhouse gases and aerosols is less well understood.

Reference 18 is missing title and journal.

This has now been corrected in the revised version.

Supp. Fig. 1. Caption mentions unit as: "(unit: m/yr)". This should either be removed or changed to the correct unit shown on the panels (i.e., m).

This has now been corrected in the revised version (see the caption in Fig. S1).

L63-64. The origin of the correlation coefficient equal to 0.53 is unclear. Is it between CESM2 and CALM? Or between CESM2 and CCI-PF?

It is the correlation coefficient between CESM2 and CCI-PF. This has now been corrected in the revised version of manuscript. (see Lines 331-332)

Lines 331-332: The CESM2-LE reasonably represents the overall spatial pattern of ALT (correlation coefficient between the CESM2-LE and CCI-PF: 0.53, $p < 0.00001$).

L64-65. The discrepancies between CESM2 and observations over (western?) Siberia are mentioned. However, the much larger discrepancies over the majority of Alaska and northwestern Canada are not mentioned, where CESM2 fails to capture large areas of discontinuous and even continuous permafrost.

This has now been corrected in the revised version (see blue revised text in main manuscript).

Lines 336-339: However, it is important to note that the model simulation underestimates the observed burned area in higher latitudes ($>60^\circ\text{N}$) (GFED4.1s: 13,081km², CESM2-LE: 2,036km²) (Figs. S2b, d). Particularly, the CESM2-LE does not well represent the observed burned area in Alaska and northwestern Canada (Figs. S2b, d).

L67-70. A quantification of the underestimation of the observed burned area over the (sub)arctic would be very valuable (e.g., observed vs. modelled burned area north of the 60th parallel). It also seems odd that Alaska has zero burned area in CESM2, while comparable areas of Canada and eastern Siberia do experience some fire.

We have added the quantification of burned area over regions above 60°N for the observation and CESM2-LE in the revised version (see blue revised text in main manuscript).

Lines 336-339: However, it is important to note that the model simulation underestimates the observed burned area in higher latitudes ($>60^\circ\text{N}$) (GFED4.1s: 13,081km², CESM2-LE: 2,036km²) (Figs. S2b, d).

Supp. Fig. 2. Panels (b) and (d) show areas north of 55 degN. All other figures with similar panels show areas north of 50 degN.

This has now been corrected in the revised version (see Fig. S2).

Supp. Fig. 3. The meaning of stippling for panels (a), (b) and (c) should be mentioned in the caption.

This has now been corrected in the revised version. (see Fig. S3).

L75. Is the word "air" correct here?

Yes. Surface air temperature is correct.

L76. If ERA5 is the source of surface temperature, it should be mentioned here in addition to CCI-PF

This has now been corrected in the revised version (see Line 64).

L84-85. The message is understandable, but the terminology used here is not correct. E.g., "outside the range" would only be appropriate if the observed value was higher than the maximum value in CESM2-LE (or lower than the minimum), which is not the case. Also, "projected changes" is usually reserved for future climate, while here the analysis focuses on the historical period.

This has now been corrected in the revised version (see blue revised text in main manuscript).

Lines 72-76: Most ensemble members in the CESM2-LE over the region show increasing trends in ALT, but the trends are on average weaker than those reconstructed from the CCI-PF (Figs. S3d-f), which suggests that the observed trends tend to be much higher than natural variability, as represented by the CESM2-LE ensemble spread.

L103. Should be "(Fig.S4)" instead of "(Fig.S3)".

This has now been corrected in the revised version.

Units of "soil moisture @ 10cm" and "soil ice @ 3m". The used units of kg/m² only make sense when these quantities are integrated over a layer of certain depth (e.g., soil moisture from 0cm to 10cm). For values at a certain depth, the correct units are kg/m³ or m³/m³. Please correct throughout the manuscript.

This has now been corrected in the revised version.

Supp. Fig. 4. There are two panels (e), and both have wrong units (division by time is missing).

This has now been corrected in the revised version.

Inconsistent definition of historical permafrost domain. For Fig. S4 (> 55 °N, ALT is less than 3m for 1850-1869). For Fig. S5 (55-60°N 60-120°W, 63-70°N 40-90°E, 60-70°N 170-180°E).

In Fig. S5, we consider the specific regions (55-60°N 60-120°W, 63-70°N 40-90°E, 60-70°N 170-180°E) to capture a clear relationship between abrupt changes in soil ice and soil moisture (and wildfire).

Fig. 2(b). Panel has incorrect units (should be mm/month).

This has now been corrected in the revised version (see Fig. 2b).

Fig. 2(e). Why anomalies relative to 1850-2100? Given that the 1850-1950 period looks stable, why not take anomalies relative to that period?

This has now been corrected in the revised version (see Fig. 2e).

What is the value of soil depth at 65.5°N, 83.75°E ? Fig. 1(h) indicates it as being close to 4m. However, Fig. 2(d-e) shows that there is moisture below 4m.

In the CESM2-LE, the value of soil depth at 65.5°N, 83.75°E is about 4.1m. The soil moisture value below 4m has been interpolated. We have revised soil depth from 5m to 3.58m in the Fig. 2 (see Figs. 2d-f).

Supp. Fig. 5. (a) Label of y-axis should be "Soil moisture @ 10 cm [Year]".

This has now been corrected in the revised version (see Fig. 4d).

Supp. Fig. 7. Units are specified as mm/yr in the caption, but the plotted values would make much more sense if the units are mm/month. Use "snowfall" instead of "snow" to avoid ambiguity. What is the difference between "(b) surface water storage runoff" and "(c) surface runoff"? Also, difference between "(f) transpiration" and "(g) plant evaporation"?

This has now been corrected in the revised version (see Fig. S5). In CLM5, surface water storage runoff represents runoff from wetlands and small, sub-grid scale water bodies. Surface water storage is parameterized as a function of the microtopography and soil moisture state of a grid cell. Transpiration is controlled by atmospheric evaporative demand through stomatal conductance.

L122-123. Typically, relative change is expressed as a fraction of the reference value, so it would be: $(51.4 - 71.1)/71.1 = -28\%$ (a 28% decrease).

This has now been corrected in the revised version (see Line 118).

L138. Use "sparse vegetation", because "reduced vegetation" suggests that vegetation is decreasing, which is the opposite of what is happening in CESM2 (Figs. S7f-g).

This has now been corrected in the revised version (see blue revised text in main manuscript).

Line 135: However, the magnitudes of abrupt changes in canopy evapotranspiration are two times smaller compared to ground evaporation changes associated with sparse vegetation across these regions (Figs. S5f-h).

L158. Is it 19.2 instead of 12.9 ?

This has now been corrected in the revised version (see Line 161).

L162-164. The sensible heat flux actually acts to cool the surface, as it transfers heat from the surface to the atmosphere. Thus, the warmer surface is the one responsible for the increased sensible heat flux, not the other way around.

Following the reviewer's suggestion, this has now been corrected in the revised version (see blue revised text in main manuscript).

Lines 166-170: Once the sensible heat flux increases abruptly following the abrupt soil drying, this can further accelerate an increase in surface air temperature, thus leading to a rapid decline in relative humidity (2040: $82.5 \pm 5.1\%$, 2060: $68.6 \pm 4.8\%$), despite a smaller change in the actual amount of water vapor (Figs. 3g-h, Figs. S8b-c).

Fig. 5(b). Please comment on the mechanisms behind the strong cooling observed over Alaska.

This has now been corrected in the revised version (see blue revised text in main manuscript).

Lines 208-210: We also observe an anomalous cooling over Alaska (Fig. 5b) may be influenced more by changes in the large-scale atmospheric circulation than by local land-atmospheric interactions.

L220-221. A word seems to be missing between "to" and "lightning".

This has now been corrected in the revised version (see blue revised text in main manuscript).

Lines 265-268: Since lightning occurs mostly in convective systems with high values of convective available potential energy, we can qualitatively assess, whether future warming is likely to increase lightning in the Arctic and Subarctic regions.

L228-236. Please include a comment on the black line in Fig.7b (GFED4.1s).

This has now been corrected in the revised version (see blue revised text in main manuscript).

Lines 283-284: However, in the CESM-LE the cumulative carbon release from wildfires tends to increase more slowly than observation (Fig.7b).

L261. Either "surface temperature" or "air temperature at 2m".

This has now been corrected in the revised version (see blue revised text in main manuscript).

Lines 311-312: 2m air temperature from the ERA5 reanalysis was used during the period 1997-2019.

L266. Missing word between "1850" and "different".

This has now been corrected in the revised version (see blue revised text in main manuscript).

Lines 315-317: We used data from the ICCP/NCAR CESM2-LE project, a single model large ensemble that was initialized in 1850 with different initial conditions based on micro and macro perturbations.

Is there a dependency between surface soil wetness and soil albedo in CLM?

Yes. In the CLM5 soil color affects the soil albedos depending on the volumetric water content of the first soil layer. The CLM5 defines 20 different soil classes, and each of which has a saturated and dry visible and infrared soil reflectance (CLM5.0 Documentation)²².

References

- 1 Teufel, B. & Sushama, L. Abrupt changes across the Arctic permafrost region endanger northern development. *Nature Climate Change* **9**, 858-862 (2019).
- 2 O'Neill, H. *et al.* Permafrost thaw and northern development. *Nature Climate Change* **10**, 722-723 (2020).
- 3 Fortier, D., Allard, M. & Shur, Y. Observation of rapid drainage system development by thermal erosion of ice wedges on Bylot Island, Canadian Arctic Archipelago. *Permafrost and Periglacial Processes* **18**, 229-243 (2007).
- 4 Arp, C. D., Drew, K. A. & Bondurant, A. C. Observation of a rapid lake-drainage event in the Arctic: Set-up and trigger mechanisms, outburst flood behaviour, and broader fluvial impacts. *Earth Surface Processes and Landforms* (2023).
- 5 Swanson, D. K. Thermokarst and precipitation drive changes in the area of lakes and ponds in the National Parks of northwestern Alaska, 1984–2018. *Arctic, Antarctic, and Alpine Research* **51**, 265-279 (2019).
- 6 Perreault, N., Lévesque, E., Fortier, D. & Lamarque, L. J. Thermo-erosion gullies boost the transition from wet to mesic tundra vegetation. *Biogeosciences* **13**, 1237-1253 (2016).
- 7 Verseghy, D. CLASS–The Canadian land surface scheme (version 3.6). *Environment Canada Science and Technology Branch Tech. Rep* **176** (2012).
- 8 Descals, A. *et al.* Unprecedented fire activity above the Arctic Circle linked to rising temperatures. *Science* **378**, 532-537 (2022).
- 9 Kharuk, V. I., Dvinskaya, M. L., Im, S. T., Golyukov, A. S. & Smith, K. T. Wildfires in the Siberian Arctic. *Fire* **5**, 106 (2022).
- 10 Webb, E. E. & Liljedahl, A. K. Diminishing lake area across the northern permafrost zone. *Nature Geoscience* **16**, 202-209 (2023).
- 11 Webb, E. E. *et al.* Permafrost thaw drives surface water decline across lake-rich regions of the Arctic. *Nature Climate Change* **12**, 841-846 (2022).
- 12 McCarter, C., Wilkinson, S., Moore, P. & Waddington, J. Ecohydrological trade-offs from multiple peatland disturbances: The interactive effects of drainage, harvesting, restoration and wildfire in a southern Ontario bog. *Journal of Hydrology* **601**, 126793 (2021).
- 13 Turetsky, M., Donahue, W. & Benscoter, B. Experimental drying intensifies burning and carbon losses in a northern peatland. *Nature communications* **2**, 514 (2011).
- 14 Seneviratne, S. I. *et al.* Investigating soil moisture–climate interactions in a changing climate: A review. *Earth-Science Reviews* **99**, 125-161 (2010).
- 15 Helbig, M. *et al.* Increasing contribution of peatlands to boreal evapotranspiration in a warming climate. *Nature Climate Change* **10**, 555-560 (2020).
- 16 Ekici, A., Lee, H., Lawrence, D. M., Swenson, S. C. & Prigent, C. Ground subsidence effects on simulating dynamic high-latitude surface inundation under permafrost thaw using CLM5. *Geoscientific Model Development* **12**, 5291-5300 (2019).
- 17 Nitzbon, J. *et al.* Fast response of cold ice-rich permafrost in northeast Siberia to a warming climate. *Nature communications* **11**, 1-11 (2020).
- 18 Dankers, R., Burke, E. & Price, J. Simulation of permafrost and seasonal thaw depth in the JULES land surface scheme. *The Cryosphere* **5**, 773-790 (2011).
- 19 Chen, Y. *et al.* Northern-high-latitude permafrost and terrestrial carbon response to two solar geoengineering scenarios. *Earth System Dynamics* **14**, 55-79 (2023).
- 20 Peng, X. *et al.* Active layer thickness and permafrost area projections for the 21st century. *Earth's Future* **11**, e2023EF003573 (2023).
- 21 Swenson, S. C., Lawrence, D. M. & Lee, H. Improved simulation of the terrestrial hydrological cycle in permafrost regions by the Community Land Model. *Journal of Advances in Modeling Earth Systems* **4** (2012).
- 22 Lawrence, D. M. *et al.* The Community Land Model version 5: Description of new features, benchmarking, and impact of forcing uncertainty. *Journal of Advances in Modeling Earth Systems* **11**, 4245-4287 (2019).

REVIEWER COMMENTS

Reviewer #1 (Remarks to the Author):

Thank you for addressing my comments. My concerns have been addressed. I do not have any major comments at this stage.

Reviewer #2 (Remarks to the Author):

Review comments are in the attached file.

Reviewer #3 (Remarks to the Author):

The authors have addressed all points raised in my first review. I have no further suggestions to enhance the quality of the manuscript.

Dear Editor and Reviewers,
Please find our responses to the reviewer's comments in blue.

Review of the revised manuscript titled "Abrupt increase in subarctic wildfires amplified by future permafrost thawing" by Kim et al.

Summary and General Comments

I appreciate the author's effort to improve the manuscript, and I would like to thank authors to reply to my comments. Unfortunately, as I understand, the author does not seem to be able to provide reasonable responses to my comments. For the reasons stated below, I cannot confirm the importance of the future abrupt permafrost thawing and increase in subarctic wildfires, which is the title of the paper. I recommend the resubmission of the paper after considering the comments below.

We thank the reviewer for the constructive comments aimed at enhancing the quality of the manuscript. Following the reviewer's comments, we have conducted additional analyses and revised the manuscript accordingly. The original reviewer comments are marked in black, and our responses are highlighted in blue text.

1. In the General Comments 1 in the review (GC1), I stated that the results were mainly based on the representative grid point and thus I couldn't understand what was happening in other places. It was nice that the authors added additional analysis (Figure S6 and S7), but the overall picture (i.e., global map) of "abrupt permafrost thawing" was not presented. Therefore, it was unclear in which areas rapid thawing of permafrost was occurring and how important it was. More specifically, it seems that the abrupt permafrost thawing would occur over a very wide range of areas from Figure 1a and 1b (the map of ALT), but at the same time the permafrost would thaw gradually (not rapidly) at some points in these regions as shown in Figure S7. Therefore, I suspect that the regions with the "abrupt changes" shown as Figure 1a-1f could include the regions with the gradual change. In the GC1, I asked that the differences between the distributions among Figure 1a-1f should be discussed (and I wanted to know where rapid thawing was occurring), but the authors did not discuss this point at all. The authors added an explanation about the analysis method (Line 357), but it is not still clear that the results possibly include the gradual change or not.

Reviewer 2 pointed out that the map for rapid changes in ALT and the regions where the rapid changes occur (which we have shown before) could potentially also include the gradual changes. To address these comments, we have therefore recalculated the rapid changes in ALT, soil ice, soil moisture, and burned area (Fig.1, 2, 4). Now, a rapid change in ALT is defined as a more than abrupt 2-fold increase in ALT over 20 years. The grid points where the ALT changes gradually over more than 20 years (see Fig.1a, b, and lines 362-368) have been excluded in our analysis and revised Figure 1. We have also updated the analysis accordingly for soil ice (Fig.1c, d, and lines 362-368). We have defined a rapid change in soil ice as an abrupt 30% decrease in total soil ice over 20 years. The revised Fig. 1 indicates that the change in ALT is more spatially consistent with the abrupt changes in soil ice and soil moisture, clearly emphasizing our main point and addressing the reviewer's concern.

We have revised the manuscript as follows; Lines 362-368: "In particular, we defined the rapid changes in ALT and subsurface runoff as a more than two-fold increase in ALT and subsurface runoff over 20 years. Grid points exhibiting a more gradual transition in ALT of more than 20 years were excluded. Similarly, we also defined the rapid changes in soil ice as a more than 30% decrease in soil ice and the rapid changes in soil moisture as a more than 20% decrease over 20 years. The rapid changes in burned area were defined as a more than two-fold increase in burned area over 20 years in the regions experiencing abrupt changes in soil ice."

Reviewer 2 further raised the question of what determines the time scale of thawing. The time scale of thawing depends on soil temperature, snow cover, and vegetation conditions etc. and varies

regionally in the model simulations.

Reviewer 2 also highlighted the necessity to address the time sequence with a regional mean. However, this approach does not well represent the rapid changes in permafrost thaw. Once the regions are aggregated for the total soil ice, the abrupt signals disappear (see Fig.R1a). This is because the spatial heterogeneity in the timing of permafrost thaw and soil ice content is high, even though the regions are close together (see Fig.R1b).

Figure R1 The timeseries in total soil ice in the 50 ensemble simulations. (a) The timeseries in regionally-averaged total soil ice (red: 60-70°N, 60-90°E, green: 50-70°N, 60-120°W, blue: 60-70°N, 150-180°E) (units: kg/m²), (b) The timeseries in soil ice over the five regions in the 50 ensemble simulations (red: 65.5°N, 80°E, yellow: 65.5°N, 82.5°E, green: 65.5°N, 85°E, blue: 65.5°N, 86.25°E, purple: 65.5°N, 87.5°E) (units: kg/m²). Thin solid lines indicate each individual simulation and thick solid lines represent the average from the 50 ensemble members.

In the same way, the authors states that “wildfires occurring in permafrost regions where the abrupt changes occurred would cumulatively release 1.40 ± 0.26 PgC towards the end of the 21st century” (Line 279). Is it really the wildfire occurred in regions with abrupt change? How did you extract the “regions where the abrupt changes occurred”? Unless the authors can extract the regions with abrupt permafrost thawing and increase in wildfire, I cannot understand the importance of these process in the Earth system. Namely, would these processes occur at only very limited points or occur over the large regions? If it is the former, these may not be an important processes in the future earth system and thus it should be discussed as it is.

Reviewer 2 also pointed out that the regions, where the rapid changes in permafrost thaw and wildfires occur together, should be considered for the Fig. 7. Following the reviewer’s comment, we have now updated the Fig. 7. We have extracted the regions where the rapid changes in total soil ice and wildfires occur together and calculated the cumulative carbon emissions from wildfires over these regions (see Fig. 7 and lines 279-284).

The revised text now reads “Lines 279-284: Our estimate from the CESM2-LE shows that wildfires occurring in permafrost regions experiencing abrupt changes would cumulatively release 322.6 ± 74.7 TgC towards the end of the 21st century and the cumulative net uptake by ecosystem production would approach about 8.9 ± 256.5 TgC in the same permafrost regions (Fig.7b). Furthermore, the contribution of carbon release from wildfires to the net carbon balance in these regions accelerates after the mid-21st century.”

Reviewer 2 suggested that if the cumulative carbon emissions from wildfires in Fig. 7 only occur at limited locations, then they may not be important processes. In our simulations, the abrupt changes in soil ice and wildfires occur markedly in western Siberia, far eastern Siberia, and Canada, with a faster-increasing rate in these regions after the mid-21st century. In addition, the net carbon release in these regions is exacerbated by the increase in wildfires at the end of the 21st century. However, as Reviewer 2 correctly pointed out, the cumulative carbon release from wildfires does not originate from entire permafrost regions, and our result accounts for a partial contribution to the future Earth system. Nevertheless, we believe that the quantitative estimation of carbon release from wildfires, and in

particular in regions with abrupt permafrost thawing, can provide an overall context on the magnitude of the projected carbon changes (see lines 284-287).

The revised manuscript now includes the following statement, line 284-287: “However, the amount of carbon released by wildfires in the regions experiencing abrupt transitions represents only a relatively small contribution to the net terrestrial carbon fluxes occurring over the entire permafrost region (north of 50°N)”.

2. In the General Comments 2 in the review (GC2), I asked the reason why the authors use the ideal experiments with 40% soil moisture reduction. In the revised manuscript, the authors describe that “The choice of 20% and 40% reduction in soil moisture for the idealized experiments is made to account for the abrupt 20-40% reduction in soil moisture following permafrost thawing in the CESM2-LE” (Line 344), but how did you calculate this number (20-40% reduction)? It is still unclear how sensitivity experiments explain future projection experiments. From Figure 5, it seems that the 20% reduction does not increase the wildfire occurrence. Are these experiments good evidence for your conclusions? As I commented in the GC2, the soil moisture reduction caused by the “abrupt” permafrost thawing would be Figure 1f. Why doesn't the author conduct a sensitivity experiment using the soil moisture changes shown in Figure 1f?

Additional questions and comments to the author's responses are as follows. Below, the original review comments are shown in gray. It seems that the author does not provide reasonable answers to the bolded and underlined parts (in GC1 and GC2).

Reviewer 2 asked for a more reasonable justification for applying the 20-40% reduction in soil moisture in the sensitivity experiment. Figure R2 shows the relative change in soil moisture (0-10cm) between 20 years pre- and post-thaw in the CESM2-LE simulations, indicating approximately 20-40% reduction in soil moisture. Based on these results, we conducted the sensitivity experiments.

Figure R2 Changes in soil moisture in 0-10cm between 20 years before and after rapid permafrost thaw over the historical permafrost regions in the CESM2-LE (units: %).

Reviewer 2 also pointed out that a 20% reduction experiment does not show an increase in the occurrence of wildfires. This is because wildfires increase logarithmically in the 40% reduction experiment compared to the 20% reduction experiment. Therefore, the figure does not visually represent the differences between the 20% and 40% experiments. To more clearly illustrate our result, we have revised the burned area unit on the logarithmic scale (see Fig. 5h). The 20% reduction experiment on the logarithmic scale shows a significant increase in wildfires relative to a control simulation and support our conclusions (see Fig. 5h).

General Comments

1. First, the causal relationships from the above 1) to 5) are shown mainly based on a “representative” grid point in northwestern Siberia (65.5N, 83.75E). Therefore, I couldn't really understand what was happening elsewhere. The authors show the maps of changes in active layer thickness, soil ice, soil moisture, runoff, evapotranspiration, Bowen ration, and burned area by performing the “change point analysis” (Figure 1-4). I am not sure that the changes in these variables are really caused by permafrost

thawing, because the map of ALT changes (Figure 1a) is not consistent with other variables (Figure 1c, 1e, 2b, 3b, 3d), and similarities and differences between these distributions are not discussed in the manuscript. In the first place, will the permafrost thaw rapidly (abruptly) at any location? If the changes in permafrost and other variables are gradual and not abrupt, what do the results of “change point analysis” mean? This analysis assumes that “one abrupt change occurs over each region” (Line 295). In fact, the distribution in Figure 1a (the timing of ALT change) appears implausible. The timing of changes in the Arctic Ocean coastline at higher latitudes is earlier than in inland areas at lower latitudes. Furthermore, the distribution of ALT appears to be uniform.

As the reviewer points out, we presented the analysis for a representative grid point in northwestern Siberia (65.5°N, 83.75°E). Following the reviewer's comment, we have included analyses for other permafrost locations (see Figs 6-7 and Lines 136-140 in main manuscript). We have also included an explanation of the similarity between ALT and deeper soil ice distributions (see Lines 89-92).

Lines 89-92: The timing of ALT and deeper pore ice tends to be similar over western Siberia and Canada. The rapid changes in ALT and deeper pore ice overall tend to occur earlier over the lower latitudes except for the regions near Arctic coastlines.

Lines 136-140: The temporal evolutions in other permafrost locations show that a rapid decrease in soil ice mainly occurs in ice-rich areas, which is consistent with sudden shifts in sub-surface runoff and soil moisture from the mid-to-end of the 21st century (Fig. S6). In contrast, in regions with less soil ice, the reduction in soil ice occurs more gradually after the early of 21st century, thereby sustaining sub-surface runoff and upper soil moisture levels (Fig. S7).

The reviewer questions whether the changes in the variables (e.g., soil moisture and runoff etc.) are really caused by permafrost thaw. Our results clearly show that significant changes in deeper soil ice are linked to the changes in upper soil moisture and burned area (Fig.1, Figs.4d-e, and Fig. S6). However, there may be inconsistencies between the changes in ALT and other variables (e.g., deeper soil ice etc.) in some regions. This discrepancy is attributed to ALT being determined by the soil temperature over the entire soil layer (0-40m) in the model, rather than focusing on hydrological properties in the deeper layer.

The reviewer points out our consideration (“one abrupt change occurs over each region”). In our study, we consider the case where only one or no abrupt transition could occur in each region in order to focus on major changes over 1850-2100 (see Lines 357-358). The change point analysis algorithm detects one structural change at each grid point during the period 1850-2100. It is therefore limited in its ability to clearly exclude some gradual changes. However, our focus is primarily on larger changes in soil ice, especially those greater than 50%, during the 20-year period between pre- and postthaw (Fig. 1d). We believe that this represents the rapid changes in permafrost thaw (Figs. 1-4, Fig. S6).

The reviewer points out that the changes in the Arctic Ocean coastline at higher latitudes occur earlier than in inland areas at lower latitudes. Although the changes in ALT occur earlier in regions close to Arctic coastlines than in inland areas at lower latitudes, the magnitude of the changes is relatively much smaller compared to other permafrost regions. In addition, the reviewer comments on the uniform spatial distribution of ALT compared to soil ice and soil moisture. This arises because ALT is determined by changes in the spatially uniform soil temperature in the model.

The authors should explain the changes in permafrost thawing and other variables should be abrupt at any location (and what determines the time scale of thawing) because they assume “one abrupt change occurs over each region” in their analysis. In addition, the readers would be interested in the time sequence at locations other than the representative grid cell. The authors should show the time sequence with regional average for different regions.

We have revised an explanation of the change point analysis (see Lines 357-358) and the analysis for the other locations in the permafrost domain. For the analysis in the other locations, we consider relatively ice-rich and ice-poor permafrost regions (see Lines 136-140 and Figs. S6-7).

Lines 357-358: To examine the largest change in each region during 1850-2100, we consider the case where only one or no abrupt change could occur in each region.

Lines 136-140: The temporal evolutions in other permafrost locations show that a rapid decrease in soil ice mainly occurs in ice-rich areas, which is consistent with sudden shifts in sub-surface runoff and soil moisture from the mid-to-end of the 21st century (Fig. S6). In contrast, in regions with less soil ice, the reduction in soil ice occurs more gradually after the early of 21st century, thereby

sustaining sub-surface runoff and upper soil moisture levels (Fig. S7).

2. I could understand the causal relationship from 1) to 4) at the representative grid cell (65.5N, 83.75E). However, the relationship between the permafrost thawing and 5) increase in wildfire occurrence is not shown with strong evidence. The author claims that wildfires increase from 2050 (Figure 4c), but it is unclear whether this is due to the permafrost thawing or changes in atmospheric conditions. In addition, the wildfire occurrence increases as shown in Figure 7b, but the role of permafrost thawing in Figure 7b is not clear (although it is the result of permafrost domain, changes in atmospheric conditions could increase the wildfire over there). The wildfire increases in the sensitivity experiments (Figure 5), but the experimental setting, i.e., abrupt 40% (80%) reductions within all layers over the high latitudes ($> 40^\circ\text{N}$), should not correspond to the conditions after permafrost thawing (the authors should explain why they chose these experimental settings).

The reviewer points out that it is unclear whether the increase in wildfires from 2050 is due to permafrost thaw or changes in atmospheric conditions. In the CESM2-LE, the abrupt increase in Arctic and Subarctic wildfires can be attributed to permafrost thaw or changes in atmospheric conditions. Therefore, to support our suggestion that the abrupt soil drying following permafrost thaw is the main driver of the abrupt increase in wildfires from the CESM2-LE, we additionally conduct the idealized experiments with perturbations of soil moisture reduction and compare them with the control simulation (see Lines 342-351 in the revised manuscript).

The control simulation represents the soil moisture state prior to permafrost thaw. In contrast, the idealized experiments are designed to include 20% and 40% soil moisture reduction perturbations at higher latitudes ($>40^\circ\text{N}$) to reflect the extent of soil moisture reduction after permafrost thaw in the CESM2-LE. The comparison between the control simulation and the soil moisture reduction experiments allows to identify the effects of abrupt soil drying on wildfires in permafrost regions. The idealized experiments clearly show that the abrupt decrease in soil moisture induces the significant atmospheric conditions, including an increase in surface air temperature and a decrease in relative humidity, that lead to an increase in wildfires (see Fig. 5).

In addition, the reviewer notes that the cumulative carbon release from wildfires in Fig. 7b can be influenced by atmospheric conditions as well as permafrost thaw. However, Fig. 7b mainly includes changes in the cumulative carbon release from wildfires following permafrost thaw. Therefore, we believe that quantifying the cumulative carbon loss from wildfires in permafrost regions, where abrupt wildfires occur, provides a rough estimate of the contribution of wildfires to the terrestrial carbon balance of these regions in a warmer climate.

Individual comments

Line 100: “Historical permafrost regions ($> 70^\circ\text{N}$)”. Do you refer to “permafrost regions” as the entire region over 70°N ? The phrase “permafrost regions” is used in the manuscript, but they are not consistent (for example, $> 55^\circ\text{N}$ in Figure 7). Why you change the region? The term “permafrost” is usually defined as the region where the ground temperature is below 0°C for at least two years. The authors should define this term and explain why they chose different regions. The definition of “historical permafrost domain” is (probably, only) shown in the caption of Supplementary Figure 4 (Line 40 in SI) as “the area where ALT is less than 3m”, but why the authors choose this threshold ($\text{ALT} < 3\text{ m}$)?

“Historical permafrost regions ($< 70^\circ\text{N}$)” refers to the areas located south of 70°N within the historical permafrost region. Our explanation in line 100 may have caused confusion, so we have corrected the text in Lines 100 and Fig. 7 and have added an explanation defining of the permafrost area in the caption of Fig. 1.

In our study, the historical permafrost region is defined as the area where ALT is less than 3m during the period of 1850-1869. Previous studies using Earth System Models (ESMs) have defined the permafrost region where ALT is less than 3m in order to focus on the near-surface permafrost¹⁻³ (Dankers, R. et al., 2011, Chen, Y. et al., 2023, Peng, X. et al., 2023, because it represents the depth range where permafrost significantly influences hydrological and biogeochemical processes.

Lines 92-94: In terms of the magnitude of the forced changes (Figs. 1b,d,f) we find a substantial increase in ALT over most of the historical permafrost regions exceeding 3m (Fig. 1b).

Now I understand that the definition of permafrost region. It would be helpful if you added it in the

main text because it is important to interpret the results.

Following the reviewer's helpful suggestions, we have now revised the main text (see lines 411-412) as follows: "Here we focus on near-surface permafrost processes. We, therefore, define the historical permafrost domain as the area where ALT is less than 3m for the period of 1850-1869".

Line 105: "abrupt increase in ALT". As shown in Supplementary Figure 3f, the trend in ALT is about 0.1 m/23yr. On the other hand, the increase in ALT shown in Figure 1h is about 40 m during about 20 years. Are these results in Supplementary Figure 3f and Figure 1h consistent? Is the ALT increase shown in Figure 1h physically reasonable (what determines the time scale of permafrost thawing)? In CLM5, the ALT is defined as the depth at which the soil temperature reaches 0°C throughout the entire soil layer in the model (0-40m). The depiction of 40m in the previous Figure 1 did not account for bedrock. Despite the model encompassing a soil depth of 40m, our study focuses on changes in near-surface permafrost. The Fig. 1h has now been updated to include bedrock considerations in the revised version (see Fig. 1h).

I cannot understand the definition of ALT. Can you really define such depth? Is it the depth below which the soil temperature become 0°C throughout the year? Why the ALT become 40m when you include the bedrock?

In the CLM5 output, the ALT is defined without consideration of the bedrock. Therefore, we have recalculated the ALT based on the soil depth from the CLM5 input file (see Fig. S4) to account for the physically meaningful change in permafrost. The modified ALT is 4.09m (see Fig.1h).

Line 107: "The timing of the 0 degC 10 cm ground temperature exceedance serves as a good proxy for the abrupt responses in soil ice and soil water", but why is the timing of "10 cm ground temperature exceeds 0 degC" so important?

The temporal evolution over Western Siberia (65.5°N, 83.75°E) shows that, after the annual mean soil temperature in 0-10cm depth reaches 0°C, delayed thawing occurs in deeper soil layer by gradual downward heat transfer (see Figs. 1-2).

This is a good explanation, so it would be nice if you added in the main text.

The explanation is included in the revised main text (see lines 120-123) "The warming propagates to deeper layers (1-3m), reaching 0°C by approximately 2050 through the downward heat transfer (Fig. 2c). Subsequently, soil ice in deeper soil layers melts away around 2050 (Fig. 2d)".

An additional question to the manuscript is as follows. I am sorry that I should give you in the first revision.

Figure 6: It would be better if you added the meaning of horizontal axis (it would be the simulation time). If it is the time, there are no tree at the left end (is this the start point of the future or historical simulation?), and trees grow during the simulation. Is this a result of simulation (i.e., the result of dynamic vegetation where trees grow after the permafrost thaw)? The meaning of tree growth should be explained.

Our simulations can represent the dynamic response of vegetation to climate conditions and CO₂ concentrations. In the simulations, vegetation increases due to CO₂ fertilization and warming in permafrost regions. This leads to an increase in fuel availability and in the frequency and intensity of wildfires. Following the reviewer's suggestion, we have updated Fig. 6 and explained the meaning of tree growth (see Fig. 6 and the Fig. 6 caption (lines 458-466)).

Lines 458-466: "Permafrost thaw occurs in response to increasing greenhouse gas concentrations when soil temperatures exceed 0°C. A rapid thaw over the ice-rich subarctic permafrost regions can trigger a subsequent abrupt drying of the upper soil due to increasing soil water percolation and an associated reduction in summer soil evaporation. This, in turn, increases sensible heat fluxes from the

surface to the atmosphere, generating near-surface atmospheric warming and an increase in atmospheric dryness. These rapidly emerging conditions can promote wildfire. Moreover, positive trends in CO₂ fertilization in the CESM-LE model further increase vegetation carbon stocks, which can serve as additional fuel for combustion, thereby contributing to the intensification of wildfires”.

REVIEWERS' COMMENTS

Reviewer #2 (Remarks to the Author):

I appreciate the author's effort to improve the manuscript, and I would like to accept the manuscript as it is.